# Learning from Mistakes: Negative Reasoning Samples Enhance Out-of-Domain Generalization

## Abstract

Supervised fine-tuning (SFT) on chain-of-thought (CoT) trajectories is a standard component of reasoning-oriented post-training for large language models. In current practice, such CoT-based SFT typically retains only trajectories whose final answers match the ground truth, which can lead to poor generalization due to overfitting and wasted data from discarding incorrect samples. Considering that incorrect samples contain implicit valid reasoning processes and diverse erroneous patterns, we investigate whether incorrect reasoning trajectories can serve as valuable supervision and surprisingly find that they substantially improve out-of-domain (OOD) generalization over correct-only training. To explain this, we performed an in-depth analysis through data, training, and inference, revealing 22 different patterns in incorrect chains, which yield two benefits: (1) *For training*, they produce a slower loss descent, indicating a broader optimization landscape that mitigates overfitting. (2) *For inference*, they raise model's policy entropy in the reasoning process by 35.67% over correct-only training (under on-policy strategy) and encourage exploration of alternative reasoning paths to improve generalization. Inspired by this, we propose **Gain-based LOss Weighting** (GLOW), an adaptive, sample-aware method that prompts models to identify underexplored patterns by rescaling sample loss weights based on inter-epoch progress. Theoretically, it converges to more generalizable solutions. Empirically, it outperforms full-data training across different model sizes and significantly improves the OOD performance of Qwen2.5-7B trained on math reasoning by 15.81% over positive-only training. Code is available at Github.

## 1 Introduction

Recent advances in large language models (LLMs), exemplified by GPT-5 (OpenAI, 2025), Gemini (Comanici et al., 2025), DeepSeek-R1 (DeepSeek-AI et al., 2025), and Qwen (Yang et al., 2025a), highlight the central role of Supervised Fine-Tuning (SFT) in modern training pipelines. By adapting base models with curated task-specific data, often enriched with Chain-of-Thougnt (CoT) annotations, SFT establishes the foundation for effective reasoning. Together with reinforcement learning (RL), which further optimizes outputs via preference-based feedback, SFT constitutes the standard two-stage paradigm underlying today's state-of-the-art LLMs. In this paradigm, we focus on the SFT stage and study the common practice of transferring reasoning ability via distilled CoT trajectories to a student model that does not initially exhibit strong reasoning behavior.

Although SFT forms the foundation of current training pipelines, existing methods remain hindered by limitations that reduce both effectiveness and efficiency, most notably two key shortcomings (Luo et al., 2024a; Chu et al., 2025; Gupta et al., 2025; Deb et al., 2025): (1) **Poor Generalization**: models tend to overfit to domain-specific reasoning shortcuts present in the demonstrations rather than learning robust, transferable reasoning capabilities (Press et al., 2022; Han et al., 2025). This often leads to performance degradation on out-of-distribution (OOD) tasks (see Tables 1 and 2 for details). (2) **Data Inefficiency**: Current reasoning post-training pipelines predominantly perform SFT on CoT trajectories distilled from a stronger teacher model, and use rejection sampling to retain only trajectories whose final answers and formats match the ground truth. Discarding samples

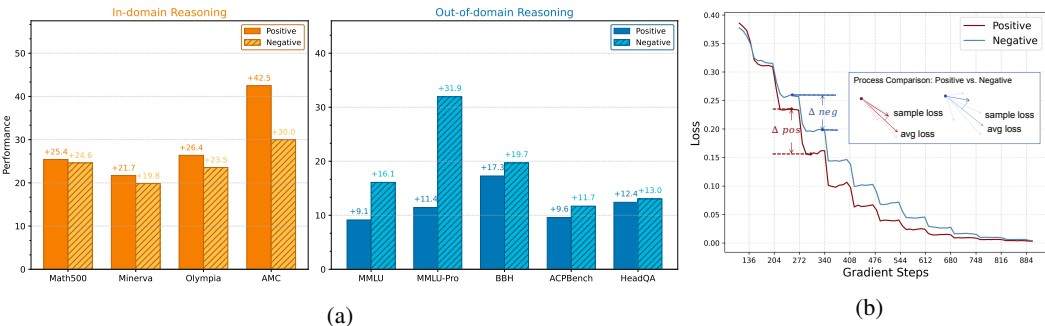

(a)             (b)

Figure 1: (a) Qwen2.5-14B trained with positive samples shows limited transfer beyond math, whereas models trained on negative samples generalize more broadly across reasoning tasks. Bars report the final accuracy on each benchmark, and the "+" annotations above them denote the absolute accuracy improvement over the model before SFT. (b) Training loss curves on MMLU for Qwen2.5-32B. The red curve corresponds to training on positive samples only, and the blue curve to training on negative samples only. Δ denotes the per-sample loss difference between epochs.

yielding incorrect answers not only wastes resources but also overlooks the correct reasoning paths potentially hidden therein (Hamdan & Yuret, 2025; Luo et al., 2024b; Li et al., 2025).

Owing to the above challenges, a natural question arises: *Are incorrect reasoning trajectories, often dismissed as noise, truly incapable of providing effective supervision?* Considering that errors encompass both valid reasoning processes and diverse erroneous patterns, these characteristics prompt us to inquire whether an SFT approach can not only improve data efficiency through utilizing all available samples but also benefit from this expanded exploration space to enhance generalization.

To investigate, we distill some data of mathematical reasoning problems and their corresponding trajectories from Qwen3-8B (Yang et al., 2025b) and split them based on whether the model's final answer matches the ground truth: correct solutions (positive) versus incorrect ones (negative). We then fine-tune a series of models, including Qwen2.5 and LLaMA (Dubey et al., 2024), on each subset separately. As illustrated in Figure 1a, the results surprisingly demonstrate that **models trained solely on negative samples outperform those trained on positive samples across many tasks, especially in the out-of-domain benchmarks.**

To understand this, we analyze it stepwise from the perspectives of data, training, and inference. The negative samples can be divided into 9 major types and 22 diverse patterns (see Table 3 and Appendix A.5), with each type serving as a distinct environment. To perform well across these environments, the model needs to learn invariant reasoning patterns, fostering better generalization. Such diversity also brings two benefits: 1) For training: loss declines more slowly than positive-only training yet converges eventually (Figure 1b), demonstrating the optimizing process for diverse reasoning patterns instead of overfitting to limited patterns. 2) For inference, models trained on negatives exhibit higher policy entropy in reasoning trajectories, indicating more diverse path exploration and boosting cross-domain generalization. Collectively, the surprising advantage of negatives over positives reveals that previously overlooked negatives can encourage the model to conduct broader, more diverse exploration during optimization, yielding more efficient, generalizable reasoning strategies.

These observations provide insights into training more generalizable models with SFT. While negative trajectories can help generalization, training on negatives alone would still be a rejection-sampling scheme that discards large portions of data. This motivates seeking a method that learns from both correct and incorrect trajectories without filtering. To address this, we introduce a dynamic mechanism called **G**ain-based **LO**ss **W**eighting (GLOW), which leverages the entire dataset without requiring prior negative sample selection. Specifically, during training, we measure a sample's value by its loss difference across consecutive epochs: a smaller difference implies minimal loss change between two optimization steps, indicating insufficient coverage of the sample's direction by other samples' optimization, and thus highlighting its greater uniqueness relative to other samples. We design a scaling function that adaptively emphasizes such samples by increasing their contribution to the loss. Theoretically, we show that this mechanism guides the model toward solutions with stronger generalization. Experimental results across models with different scales demonstrate its consistent improvements. In particular, on Qwen2.5-7B, GLOW achieves an average improvement

of 2.14% over mixed-data training and gains of 5.51% in OOD scenarios compared to training with only positive samples.

In all, our core contributions can be summarized as follows:

- We provide the first systematic study demonstrating that negative reasoning samples constitute valuable supervision: fine-tuning on them improves out-of-distribution generalization. This offers a novel perspective on mitigating overfitting in SFT by exploiting these data.
- We provide a deep analysis of how negatives improve generalization from data, training, and inference perspectives, which reflects that negative samples can enable the model to conduct broader exploration of reasoning paths and directly strengthen generalization.
- We propose a novel GLOW mechanism that adaptively recognize and amplifies the contribution of samples with the highest training gain, measured by their loss reduction trajectory. This approach improves the utility of negative samples, enhances generalization, and offers a practical path toward more data-efficient SFT.

## 2 RELATED WORKS

**Supervised Fine-Tuning for Reasoning** SFT has emerged as a central approach for improving the reasoning capabilities of large language models (Wei et al., 2021; Ouyang et al., 2022). It adapts a general-purpose model to downstream tasks or desired behaviors by training on carefully curated datasets. To ensure data quality, rejection sampling (Ahn et al., 2024) is often employed as a filtering strategy that discards samples failing to meet predefined standards. Recent studies further show that SFT can transfer long CoT reasoning patterns from larger models to smaller ones (Shao et al., 2024a; Zheng et al., 2025; Yu et al., 2025b), thereby enhancing the reasoning performance of resource-efficient models. In addition, SFT provides a strong initialization for reinforcement learning by aligning models with human-preferred behaviors before optimization (Lewkowycz et al., 2022; Shao et al., 2024b). However, this reliance on heavily filtered data inevitably wastes data, as a large portion of available supervision is discarded.

**Learning from Negative Data** Learning from negative samples can be broadly grouped into prompt-based, fine-tuning–based, and reinforcement-learning–based approaches. Prompt-based methods use negative examples to steer model behavior. Gao & Das (2024) employ them to encode ambiguous preferences that models should avoid, while Alazraki et al. (2025) show that inserting a negative example into the prompt can be more effective than adding an additional positive one, and that providing incorrect rationales may even over-constrain the model. However, the effectiveness of such methods is ultimately limited by the model's own reasoning and instruction-following abilities. By contrast, fine-tuning–based approaches are more commonly used to strengthen reasoning or to provide a strong initialization for subsequent reinforcement learning (Guo et al., 2025). Some studies distill positive CoT trajectories from initially negative samples using teacher models (Yu et al., 2025a; Pan et al., 2025; An et al., 2023), whereas others introduce explicit prefixes to distinguish positive from negative samples (Wang et al., 2024a; Tong et al., 2024). Beyond SFT, recent reinforcement-learning (RL) methods for reasoning language models also explore how to exploit negative signals. Examples include decomposing RL with verifiable rewards into separate positive and negative reinforcement (Zhu et al., 2025), reactivating residual prompts through exploration (Liu et al., 2025), mining useful steps within otherwise incorrect trajectories (Yang et al., 2025d), and converting homogeneous errors into informative gradients (Nan et al., 2025). Nevertheless, in both SFT and RL, negative samples are still typically treated as less valuable than positive ones and are used mainly as penalties, down-weighted rewards, or auxiliary signals.

**Domain Generalization in LLMs** Most fine-tuning studies prioritize improving reasoning within a single domain such as mathematics or code, while systematic treatment of cross-domain transfer remains limited. For example, Huan et al. (2025) study math data and show that SFT induces significant latent space and token rank shifts, which lead to forgetting of general capabilities. Wu et al. (2025) introduce two metrics, knowledge index and information gain, to disentangle knowledge from reasoning, finding that SFT on math provides little benefit in knowledge-intensive domains such as medicine. Similarly, Yang et al. (2025c) and Zhao et al. (2025) argue that SFT often constructs only superficial reasoning chains and fails to transfer effectively across domains. However,

these studies are primarily diagnostic analyses: they do not propose concrete methods, nor do they investigate the problem from a data-centric perspective.

## 3 NEGATIVE SAMPLES ENHANCE OUT-OF-DOMAIN REASONING

In this section, we describe the empirical phenomenon that motivates our study: fine-tuning on negative reasoning samples can enhance OOD generalization more effectively than fine-tuning on positive samples. We first detail the controlled experiments designed to validate this phenomenon and then present results that demonstrate its consistency across diverse benchmarks and model scales.

### 3.1 DATA CONSTRUCTION AND TRAINING SETUP

We use Qwen3-8B to distill responses from OpenMathReasoning (Moshkov et al., 2025) and the MMLU (Hendrycks et al., 2021b) training set as training data for mathematical and general reasoning tasks. Responses that matched the final answer are classified as positive, while others are defined as negative. To ensure a fair comparison, we sample an equal number of positive and negative responses, each containing the complete reasoning format. We then use Qwen-2.5 series (3B, 7B, 14B, and 32B) model and Llama-3.1 8B for SFT training. For more detailed training configurations, please refer to the Appendix 3.1.

### 3.2 NEGATIVES SURPASS POSITIVES IN OUT-OF-DOMAIN

Table 1: Cross-domain performance of models trained on the **math reasoning** dataset. "Avg." denotes the average score within each group. Colored cells highlight entries that support our findings: orange cells mark in-domain benchmarks where positives outperform negatives, and blue cells mark out-of-domain benchmarks where negatives outperform positives. Within each positive/negative pair, the higher score is additionally highlighted in the corresponding color.

| Model | Setting | Math Reasoning (In-Domain) | | | | | General Reasoning (Out-of-Domain) | | | | Other Reasoning (Out-of-Domain) | | |
|---|---|---|---|---|---|---|---|---|---|---|---|---|---|
| | | Math500 | Minerva | Olympia | AMC | Avg. | MMLU | MMLU-Pro | BBH | Avg. | ACPBench | HeadQA | Avg. |
| Qwen2.5-3B | Base | 52.60 | 21.32 | 22.52 | 32.50 | 32.24 | 31.88 | 12.54 | 27.75 | 24.06 | 23.31 | 33.15 | 28.23 |
| | Full | 60.80 | 26.10 | 23.26 | 35.00 | 36.29 | 64.13 | 38.66 | 52.29 | 51.69 | 32.68 | 62.69 | 47.69 |
| | Positive | 61.60 | 25.74 | 24.44 | 42.50 | 38.60 | 54.45 | 25.62 | 44.35 | 41.50 | 30.21 | 59.81 | 45.01 |
| | Negative | 58.60 | 23.53 | 24.15 | 42.50 | 37.20 | 64.09 | 39.20 | 53.87 | 52.39 | 33.06 | 63.13 | 48.10 |
| | Δ(pos-neg) | +3.00 | +2.21 | +0.29 | 0.00 | +1.38 | -9.64 | -13.58 | -9.52 | -10.91 | -2.85 | -3.32 | -3.09 |
| Qwen2.5-7B | Base | 58.40 | 26.84 | 26.07 | 52.50 | 40.95 | 55.80 | 26.56 | 51.10 | 44.49 | 28.77 | 57.29 | 43.03 |
| | Full | 76.60 | 40.07 | 38.96 | 55.00 | 52.66 | 72.24 | 53.71 | 70.84 | 65.60 | 38.27 | 72.06 | 55.17 |
| | Positive | 78.00 | 36.76 | 41.78 | 57.50 | 53.51 | 61.03 | 32.70 | 60.58 | 51.44 | 33.38 | 68.60 | 50.99 |
| | Negative | 77.60 | 40.44 | 38.37 | 57.50 | 53.48 | 73.11 | 53.74 | 71.73 | 66.19 | 38.98 | 71.81 | 55.40 |
| | Δ(pos-neg) | +0.40 | -3.68 | +3.41 | 0.00 | +0.03 | -12.08 | -21.04 | -11.15 | -14.76 | -5.60 | -3.21 | -4.41 |
| Qwen2.5-14B | Base | 62.60 | 26.84 | 27.56 | 40.00 | 39.25 | 64.68 | 35.77 | 59.27 | 53.24 | 37.04 | 68.75 | 52.90 |
| | Full | 86.80 | 47.79 | 52.30 | 82.50 | 67.35 | 81.56 | 67.63 | 80.90 | 76.70 | 48.13 | 81.44 | 64.79 |
| | Positive | 88.00 | 48.53 | 53.93 | 82.50 | 68.24 | 73.81 | 47.21 | 76.54 | 65.85 | 46.62 | 81.15 | 63.89 |
| | Negative | 87.20 | 46.69 | 51.11 | 70.00 | 63.75 | 80.77 | 67.70 | 78.95 | 75.81 | 48.73 | 81.77 | 65.25 |
| | Δ(pos-neg) | +0.80 | +1.84 | +2.82 | +12.50 | +4.49 | -6.96 | -20.49 | -2.41 | -9.95 | -2.11 | -0.62 | -1.37 |
| Qwen2.5-32B | Base | 63.20 | 34.19 | 26.52 | 35.00 | 39.73 | 68.34 | 39.80 | 58.65 | 55.60 | 38.63 | 68.45 | 53.54 |
| | Full | 92.20 | 52.57 | 57.19 | 85.00 | 71.74 | 85.22 | 73.10 | 83.53 | 80.62 | 50.67 | 84.90 | 67.79 |
| | Positive | 91.40 | 50.74 | 60.89 | 85.00 | 72.01 | 79.01 | 54.31 | 80.61 | 71.31 | 49.96 | 83.15 | 66.56 |
| | Negative | 92.20 | 50.74 | 58.37 | 95.00 | 74.08 | 85.47 | 73.53 | 84.51 | 81.17 | 51.80 | 85.27 | 68.54 |
| | Δ(pos-neg) | -0.80 | 0.00 | +2.52 | -10.00 | -2.07 | -6.46 | -19.22 | -3.90 | -9.86 | -1.84 | -2.12 | -1.98 |
| Llama3.1-8B | Base | 2.80 | 1.10 | 0.44 | 0.00 | 1.09 | 66.49 | 0.47 | 2.33 | 23.10 | 5.18 | 2.30 | 3.74 |
| | Full | 41.20 | 18.01 | 14.67 | 15.00 | 22.22 | 62.48 | 36.88 | 55.12 | 51.49 | 32.96 | 65.90 | 49.43 |
| | Positive | 37.80 | 18.01 | 10.37 | 12.50 | 19.67 | 41.95 | 23.15 | 45.07 | 36.72 | 31.20 | 47.81 | 39.50 |
| | Negative | 34.40 | 18.38 | 9.19 | 20.00 | 20.49 | 62.14 | 36.22 | 54.85 | 51.07 | 33.31 | 65.17 | 49.24 |
| | Δ(pos-neg) | +3.40 | -0.37 | +1.18 | -7.50 | -0.82 | -20.19 | -13.07 | -9.78 | -14.35 | -2.11 | -17.36 | -9.74 |

As shown in Table 1 and Table 2, we surprisingly find that training on negative samples, although it yields smaller improvements than positive samples on in-domain performance, consistently consistently improves OOD generalization. Overall, models trained on negative math reasoning samples achieve an average improvement of 11.97% on general reasoning tasks and 4.11% on other reasoning tasks. Similarly, models trained on negative MMLU samples gain an average of 1.98% on mathematical reasoning and 1.35% on other reasoning benchmarks. Although mathematical problems are generally more suitable for constructing reasoning-focused data, the same trend is observed for models trained on MMLU, indicating that the benefit of negative samples for OOD generalization is not limited to a specific domain. These observations motivate a deeper analysis into the underlying factors that make negative samples more effective for enhancing OOD reasoning performance.

Table 2: Cross-domain performance of models trained on the **general reasoning** dataset. "Avg." denotes the average score within each group. Colored cells highlight entries that support our findings: orange cells mark in-domain benchmarks where positives outperform negatives, and blue cells mark out-of-domain benchmarks where negatives outperform positives. Within each positive/negative pair, the higher score is additionally highlighted in the corresponding color.

| Model | Setting | Math Reasoning (Out-of-Domain) | | | | | General Reasoning (In-Domain) | | | | Other Reasoning (Out-of-Domain) | | |
|---|---|---|---|---|---|---|---|---|---|---|---|---|---|
| | | Math500 | Minerva | Olympia | AMC | Avg. | MMLU | MMLU-Pro | BBH | Avg. | ACPBench | HeadQA | Avg. |
| Qwen2.5-3B | Base | 52.60 | 21.32 | 22.52 | 32.50 | 32.24 | 31.88 | 12.54 | 27.75 | 24.06 | 23.31 | 33.15 | 28.23 |
| | Full | 58.20 | 23.16 | 25.19 | 35.00 | 35.39 | 66.74 | 40.82 | 53.35 | 53.64 | 35.70 | 67.61 | 51.66 |
| | Positive | 59.20 | 27.21 | 25.04 | 30.00 | 35.36 | 67.88 | 42.56 | 52.84 | 54.43 | 34.93 | 67.69 | 51.31 |
| | Negative | 59.60 | 28.31 | 25.48 | 40.00 | 38.35 | 65.42 | 38.55 | 52.28 | 52.08 | 36.13 | 68.85 | 52.49 |
| | $\Delta$(pos-neg) | -0.40 | -1.10 | -0.44 | -10.00 | -2.99 | +2.32 | +4.01 | +0.56 | +2.30 | -1.20 | -1.16 | -1.18 |
| Qwen2.5-7B | Base | 58.40 | 26.84 | 26.07 | 52.50 | 40.95 | 55.80 | 26.56 | 51.10 | 44.49 | 28.77 | 57.29 | 43.03 |
| | Full | 75.60 | 38.60 | 40.15 | 47.50 | 50.46 | 73.14 | 51.15 | 71.30 | 65.20 | 42.18 | 72.76 | 57.47 |
| | Positive | 74.40 | 37.50 | 39.85 | 50.00 | 50.44 | 73.42 | 53.22 | 68.23 | 64.96 | 40.32 | 74.25 | 57.29 |
| | Negative | 77.00 | 37.13 | 42.07 | 60.00 | 54.05 | 71.23 | 45.79 | 69.46 | 62.16 | 42.61 | 73.38 | 58.00 |
| | $\Delta$(pos-neg) | -2.60 | +0.37 | -2.22 | -10.00 | -3.61 | +2.19 | +7.43 | -1.23 | +2.80 | -2.29 | +0.87 | -0.71 |
| Qwen2.5-14B | Base | 62.60 | 26.84 | 27.56 | 40.00 | 39.25 | 64.68 | 35.77 | 59.27 | 53.24 | 37.04 | 68.75 | 52.90 |
| | Full | 82.20 | 43.01 | 51.85 | 70.00 | 61.77 | 78.13 | 59.57 | 80.56 | 72.75 | 48.87 | 79.94 | 64.41 |
| | Positive | 80.20 | 42.28 | 50.96 | 72.50 | 61.49 | 80.09 | 65.26 | 80.21 | 75.19 | 48.56 | 80.53 | 64.55 |
| | Negative | 83.00 | 45.22 | 48.89 | 65.00 | 60.53 | 76.83 | 56.03 | 80.15 | 71.00 | 48.27 | 80.56 | 64.42 |
| | $\Delta$(pos-neg) | -2.80 | -2.94 | +2.07 | +7.50 | +0.96 | +3.26 | +9.23 | +0.06 | +4.18 | +0.29 | -0.03 | +0.13 |
| Qwen2.5-32B | Base | 63.20 | 34.19 | 26.52 | 35.00 | 39.73 | 68.34 | 39.80 | 58.65 | 55.60 | 38.63 | 68.45 | 53.54 |
| | Full | 86.60 | 46.69 | 55.70 | 80.00 | 67.25 | 79.06 | 61.15 | 79.94 | 73.38 | 49.89 | 83.01 | 66.45 |
| | Positive | 85.20 | 46.69 | 56.15 | 75.00 | 65.76 | 81.97 | 68.54 | 81.60 | 77.37 | 50.35 | 82.90 | 66.63 |
| | Negative | 86.40 | 47.06 | 56.89 | 72.50 | 65.71 | 77.99 | 58.34 | 80.71 | 72.35 | 51.20 | 82.39 | 66.80 |
| | $\Delta$(pos-neg) | -1.20 | -0.37 | -0.74 | +2.50 | +0.05 | +3.98 | +10.20 | +0.89 | +5.02 | -0.85 | +0.51 | -0.17 |
| Llama3.1-8B | Base | 2.80 | 1.10 | 0.44 | 0.00 | 1.09 | 66.49 | 0.47 | 2.33 | 23.10 | 5.18 | 2.30 | 3.74 |
| | Full | 20.00 | 15.81 | 6.52 | 2.50 | 11.21 | 66.49 | 40.56 | 53.73 | 53.59 | 36.06 | 69.55 | 52.81 |
| | Positive | 15.60 | 11.76 | 3.85 | 7.50 | 9.68 | 64.73 | 39.74 | 45.39 | 49.95 | 29.61 | 67.69 | 48.65 |
| | Negative | 23.00 | 16.18 | 6.67 | 10.00 | 13.96 | 64.63 | 38.85 | 53.23 | 52.24 | 37.15 | 69.80 | 53.48 |
| | $\Delta$(pos-neg) | -7.40 | -4.42 | -2.82 | -2.50 | -4.29 | +0.10 | +0.89 | -7.84 | -2.28 | -7.54 | -2.11 | -4.83 |

# 4 WHY NEGATIVE IS BETTER

To better understand why negative samples benefit out-of-distribution generalization, we analyze this phenomenon step by step. Empirically, we observe that correct trajectories usually share a few common success factors (such as accurate computation and proper problem understanding), whereas the reasons for failure are much more diverse. We therefore first examine the data to characterize how negative samples introduce greater diversity. We then study the training dynamics to reveal how this diversity influences optimization. Finally, we analyze model behaviors during inference to show how these training effects translate into stronger generalization. This step-by-step analysis sheds light on the mechanism through which negatives improve OOD performance.

## 4.1 DATA PERSPECTIVE

Following (He et al., 2025), we categorize reasoning errors into 9 major types and 22 subtypes. For each negative sample in the OpenMathReasoning and MMLU training datasets, we employ Gemini-2.5-Pro (Comanici et al., 2025) to assign its error category (see Appendix A.5 for the prompt used). As shown in Table 3, the distribution of error types is highly diverse, covering a wide spectrum from logical errors to comprehension errors. Negative samples exhibit a richer variety of reasoning patterns, whereas positive data tend to follow more consistent trajectories. Detailed classification can be found in Appendix A.3.

Table 3: Error categorization in the negative OpenMathReasoning and MMLU samples.

| Error Categories | OpenMathReasoning | MMLU |
|---|---|---|
| Calculation | 27 | 9 |
| Completeness | 11 | 28 |
| Evaluation System | 2599 | 2024 |
| Formal | 57 | 123 |
| Knowledge | 27 | 199 |
| Logical | 195 | 4116 |
| Programming | 8 | 5 |
| Understanding | 435 | 1056 |
| Special Cases | 301 | 1137 |
| **Total** | **3660** | **8697** |

This phenomenon can be understood more formally through the lens of Invariant Risk Minimization (IRM) (Arjovsky et al., 2019). IRM posits that generalization improves when a model learns representations that capture invariant causal structure across diverse environments. In our setting, we interpret different categories of incorrect reasoning as different environments: each error type induces its own sub-distribution over inputs and outputs, with characteristic failure patterns that de-

fine a distinct local data distribution. Importantly, many negative samples still contain partially valid reasoning paths, as illustrated in Figure 8, so these environments are far from pure noise. Exposure to many such environments requires the model to perform well under varied failure modes, which in turn encourages it to learn reasoning features that remain stable across them.

Formally, let $\mathcal{E}$ denote the set of environments induced by these negative error categories, and let each $e \in \mathcal{E}$ correspond to a data distribution $D^e$ over input–output sequences $(x, y)$. We decompose the language model into a shared sequence representation $\Phi$ and a shared next-token predictor $w$, where $\Phi$ represents the main layers of the model and $w$ is the vocabulary projection head. IRM in the autoregressive setting requires that the same predictor $w$ be optimal across all environments when paired with $\Phi$:

$$\min_{\Phi} \sum_{e \in \mathcal{E}} R^e(w \circ \Phi) \quad \text{subject to} \quad w \in \arg\min_{w'} R^e(w' \circ \Phi), \ \forall e \in \mathcal{E}, \tag{1}$$

where the per-environment autoregressive risk is

$$R^e(w \circ \Phi) = \mathbb{E}_{(x,y) \sim D^e} \left[ \sum_{t=1}^{|y|} \ell\big(w(\Phi(x, y_{<t})), y_t\big) \right], \tag{2}$$

and $\ell$ denotes the cross-entropy loss. Because the predictor $w$ is shared across all environments, achieving optimality requires the representation $\Phi$ to encode reasoning features that remain reliable under different types of errors. This provides a conceptual explanation of why diverse negative samples, which span many environments, can improve the robustness and out-of-distribution generalization of the learned reasoning patterns.

From this perspective, positive samples are clean and correct but occupy a relatively narrow range of environments, which limits their ability to support invariance. Negative samples, in contrast, cover multiple environments and expose diverse failure modes within otherwise valid reasoning structures. This diversity pushes the model to learn more robust representations that generalize across heterogeneous reasoning scenarios.

## 4.2 TRAINING PERSPECTIVE

To characterize the learning dynamics, we plot the training loss every 10 steps for all models fine-tuned on positive and negative samples from math reasoning and MMLU. We present Qwen2.5-32B (Figure 1b) as a representative example, while others are provided in Appendix A.8. The curves follow a consistent stage-wise pattern. Loss drops sharply at the end of each epoch for positive samples, leading to faster initial convergence, whereas negative samples produce a smoother,

Table 4: Comparison of training dynamics of Qwen2.5-32B under positive and negative MMLU settings. Each value represents the **difference** between the per-epoch loss drops of the Positive ($\Delta_{\text{pos}}$) and Negative ($\Delta_{\text{neg}}$), i.e., $\Delta_{\text{pos}} - \Delta_{\text{neg}}$. Small decimal values are expected, and the interpretation relies on the relative difference.

| Model | $\Delta_{\text{avg\_loss}}^{\text{epoch } 2-1}$ | $\Delta_{\text{avg\_loss}}^{\text{epoch } 3-2}$ | $\Delta_{\text{avg\_loss}}^{\text{epoch } 4-3}$ | $\Delta_{\text{avg\_loss}}^{\text{epoch } 5-4}$ |
|---|---|---|---|---|
| Qwen2.5-3B | 0.014957 | 0.013486 | 0.015686 | 0.014000 |
| Qwen2.5-7B | 0.009729 | 0.022514 | 0.014172 | 0.001156 |
| Qwen2.5-14B | 0.008515 | 0.017786 | 0.011157 | 0.005472 |
| Qwen2.5-32B | 0.007143 | 0.018200 | 0.015557 | 0.003772 |
| Llama3.1-8B | 0.015586 | 0.023344 | 0.005571 | 0.004915 |

gradual decline that ultimately reaches a comparable loss floor. This is because the optimization directions of individual samples align more consistently with the average gradient in the positive set than in the negative set, while negative samples point to a wider exploration space. We quantify this behavior using the average loss difference between consecutive epochs, as reported in Table 4, which confirms that positives decrease faster in the early stages. The negative-trajectory loss decreases consistently throughout training and follows a trend similar to the positives (see Figure 1b and Figure 9). More importantly, this is accompanied by steady improvements on held-out evaluations at 5/10/15 epochs (see Table 9 and appendix A.9), suggesting that negatives act as learnable supervision rather than pure noise. They encode diverse exploratory patterns where incorrect answers coexist with partially valid reasoning, offering sustained constraints that encourage the model to develop more robust reasoning strategies instead of memorizing a single correct trajectory.

These results show that the value of negative samples lies in their diversity. Although this slows loss reduction by introducing varied optimization directions, it compels the model to explore a broader reasoning space and converge to more generalizable patterns.

## 4.3 INFERENCE PERSPECTIVE

After analyzing the properties of the training data and the characteristics of the optimization process, we further investigate what drives the superior out-of-distribution performance of models trained with negatives. To this end, we focus on policy entropy, which provides a principled measure of the uncertainty and exploration in model reasoning. We investigate how training on different types of trajectories shapes the entropy dynamics of model reasoning. We first analyze the policy entropy of the model. We use $M_{pos}$ to denote the model trained on the positive subset of OpenMathReasoning, and $M_{neg}$ for the one trained on the negative subset. To assess entropy in both in-domain and out-of-domain settings, we distill trajectories with reasoning trace and final answers from Qwen3-8B on a math set (denoted as "Math") and an OOD set (denoted as "Other").

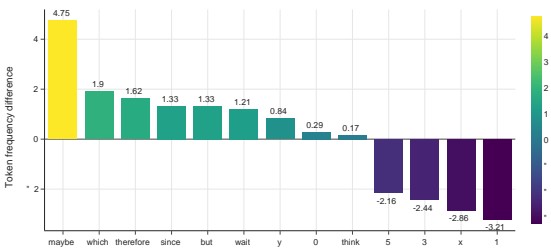

Figure 2: Token frequency differences between $M_{neg}$ and $M_{pos}$ on digits and high-entropy tokens.

Table 5: Policy entropy analysis on $M_{pos}$ and $M_{neg}$.

| Model | Setting | Data | $\bar{H}_{\text{think}}$ | $\bar{H}_{\text{ans}}$ | $\Delta H$ |
|---|---|---|---|---|---|
| $M_{pos}$ | Off-policy | Math | 0.909 | 0.708 | 0.202 |
| | | Other | 1.138 | 0.873 | 0.265 |
| | On-policy | Math | 0.753 | 0.601 | 0.153 |
| | | Other | 0.669 | 0.757 | -0.088 |
| $M_{neg}$ | Off-policy | Math | 1.212 | 0.883 | 0.329 |
| | | Other | 1.427 | 0.992 | 0.435 |
| | On-policy | Math | 1.011 | 0.772 | 0.239 |
| | | Other | 0.917 | 0.783 | 0.134 |

We mark the thinking span as the tokens between `<think>` and `</think>`, and the answer span as the tokens after `</think>`. For each prompt $x$, we compute the policy entropy within these spans under two rules: (i) **off-policy**: measuring under the teacher's reference trajectory; and (ii) **on-policy**: the model generates its own trajectory under a fixed decoding rule. Unless noted, entropy is computed from raw $T{=}1$ logits (no temperature rescaling), not excluding padding and special boundary tokens.

Formally, let $\mathcal{V}$ be the vocabulary and $\theta$ the model parameters. At step $t$ the token-level policy entropy is

$$H_t(\theta \mid x, y_{<t}) = -\sum_{v \in \mathcal{V}} p_\theta(v \mid x, y_{<t}) \log p_\theta(v \mid x, y_{<t}), \tag{3}$$

where $p_\theta(\cdot \mid x, y_{<t})$ is induced by pre-softmax logits. For each sample $i$, let $\mathcal{T}_{\text{think}}^{(i)}$ and $\mathcal{T}_{\text{ans}}^{(i)}$ denote the token within thinking and answer spans, respectively. The spans are determined by model's own generation (on-policy) or teacher's trajectory (off-policy). We report the average entropy per span:

$$\bar{H}_{\text{think}}^{(i)} = \frac{1}{\left|\mathcal{T}_{\text{think}}^{(i)}\right|} \sum_{t \in \mathcal{T}_{\text{think}}^{(i)}} H_t, \qquad \bar{H}_{\text{ans}}^{(i)} = \frac{1}{\left|\mathcal{T}_{\text{ans}}^{(i)}\right|} \sum_{t \in \mathcal{T}_{\text{ans}}^{(i)}} H_t, \tag{4}$$

as well as the entropy drop across the boundary:

$$\Delta H^{(i)} = \bar{H}_{\text{think}}^{(i)} - \bar{H}_{\text{ans}}^{(i)}, \tag{5}$$

Results in Table 5 show that models trained on negative trajectories sustain higher policy entropy in thinking span and exhibit a larger boundary gap, **indicating broader search followed by sharper commitment and aligning with their stronger cross-domain transfer.** Moreover, Off-policy are consistently higher than on-policy, since teacher-forcing trajectories push the model into low-confidence neighborhoods with diffuse distributions, while self-decoding remains confined to a few high-confidence modes that yield lower entropy. Under distribution shift, entropy increases for both spans. The positive-trained model degrades most and even flips the margin on on-policy OOD, indicating unstable calibration and over-specialization to in-domain templates. Overall, negative supervision induces a "high-entropy reasoning" profile that better predicts generalization.

We then analyze the distribution of high-entropy tokens in the trajectories generated by different models. Figure 2 shows the token frequency distribution difference per trajectory for $M_{pos}$ and $M_{neg}$. Compared with $M_{\text{pos}}$, $M_{neg}$ produces substantially more discourse and hesitation tokens such as "maybe," "wait," and "but," while emitting numerals less frequently, indicating that its trajectories devote more budget to exploratory connective reasoning than committing to numeric content. We also visualize the trajectories generated by the two models in Figure 7, showing that during inference, $M_{neg}$ has a higher effective branching factor, enabling the model to maintain multiple continuations plausible and to explore more reasoning paths before committing to an answer.

## 5 BETTER LEVERAGING OF NEGATIVE

In this section, we move beyond the empirical finding that negatives improve out-of-distribution generalization. Relying solely on negatives is essentially a form of rejection sampling and does not make efficient use of the data, as sample quality cannot be determined simply by correctness. Our goal is to develop models that achieve strong performance on both in-domain and out-of-distribution settings with higher data efficiency. To this end, we focus on the training process as the most principled direction for improvement. Building on the analysis of training dynamics, we present a general mechanism, establish its theoretical foundation, and validate its effectiveness through experiments.

### 5.1 GAIN-BASED LOSS WEIGHTING

Negative trajectory supervision improves reasoning because it enlarges the model's effective training space. This is evidenced by three key observations: (1) compared to positive training, negative training yields similarly shaped learning curves but slower convergence at a fixed step budget, indicating that updates are less concentrated along a few dominant directions and thus avoid early collapse into limited reasoning patterns. (2) Analysis in Section 4.3 shows that models trained with negatives exhibit a higher policy entropy, thereby gaining a greater capacity for exploration. Taken together, our observations suggest a practical motivation: reweight the objective to amplify the loss contributions of under-explored samples, dynamically steering updates toward complementary directions and yielding progressively larger incremental gains.

We use $\ell_i^{(t)}$ to denote the loss of sample $i$ at epoch $t$. We assess the learning progress of each sample by the reduction in its loss across consecutive epochs. Samples with small loss reductions correspond to patterns that remain insufficiently learned and offer higher optimization utility, while large reductions imply saturated learning with limited marginal utility. We therefore use $\Delta_i^{(t)} = \ell_i^{(t-1)} - \ell_i^{(t)}$ to identify under-learned samples and amplify their impact, ensuring that training prioritizes the regions where the model can still achieve the greatest gain. Specifically, the contribution of each sample is adjusted according to $\Delta_i^{(t)}$:

$$w_i^{(t)} = \alpha\big(1 - \sigma\big(\beta\Delta_i^{(t)}\big)\big), \tag{6}$$

where $\sigma(\cdot)$ is the sigmoid function, and $\alpha, \beta$ are scaling hyperparameters. For the first epoch, we set $w_i^{(1)} = 1$ for all samples. The reweighted training objective becomes:

$$\mathcal{L}_{\text{GLOW}}^{(t)}(\theta) = \sum_{i=1}^{N} w_i^{(t)} \cdot \ell_i^{(t)}. \tag{7}$$

**Theoretical View**  We provide a sketch of why the reweighted objective in Eq. 7 improves generalization. Consider one gradient update at step $t$, $\theta^{(t)} = \theta^{(t-1)} - \eta G^{(t-1)}$ with $G^{(t-1)} = \sum_i w_i^{(t-1)} g_i^{(t-1)}$. By the $L$-smoothness of the loss, a Taylor expansion gives

$$\Delta_i^{(t)} = \ell_i^{(t-1)} - \ell_i^{(t)} \approx \eta\, g_i^{(t-1)\top} G^{(t-1)} - \tfrac{1}{2}\eta^2 G^{(t-1)\top} H_i G^{(t-1)}, \tag{8}$$

where $H_i$ is the Hessian of model parameters. The leading term shows that $\Delta_i^{(t)}$ is large if $g_i^{(t-1)}$ aligns with the dominant descent directions $G^{(t-1)}$, and small otherwise. Hence, Eq. 6 adaptively increases the weight of samples whose gradients lie in less explored directions.

Let $F_w = \frac{1}{N} \sum_i w_i^{(t)} g_i^{(t)} g_i^{(t)\top}$ denote the empirical Fisher, which quantifies the extent of the model's directional exploration in parameter space. Increasing $w_i^{(t)}$ for small-$\Delta_i^{(t)}$ samples adds positive semi-definite increments $\Delta w_i g_i g_i^\top$ along diverse directions. By Weyl's inequality (Weyl, 1912), this raises the smaller eigenvalues of $F_w$, improving its effective rank and conditioning (Horn & Johnson, 2012). Since $F_w$ approximates the Hessian in standard settings (Martens, 2020), the optimization landscape becomes better conditioned, leading to more balanced descent across directions. Stability-based generalization bounds (Bousquet & Elisseeff, 2002; Hardt et al., 2016) then imply a tighter generalization bound, as flatter and more isotropic minima correlate with improved robustness (Keskar et al., 2016; Neyshabur et al., 2017).

In summary, the dynamic weighting in Eq. 6 systematically enlarges gradients from diverse, less-explored reasoning trajectories (often negatives), increases gradient diversity, and thus improves both optimization and generalization. For detailed proof, see Appendix A.2.

## 5.2 EXPERIMENTAL RESULTS

Building on the theoretical analysis, we empirically validate the effectiveness of GLOW in the SFT stage. All other experimental settings are the same as 3.1 and details are described in Appendix A.1.

Table 6: Cross-domain performance of models trained on the **math reasoning** dataset. "Avg." denotes the average score within each group. **Bold** indicates the best results under the same model.

| Model | Setting | Math Reasoning (In-Domain) | | | | | General Reasoning (Out-of-Domain) | | | | Other Reasoning (Out-of-Domain) | | |
| --- | --- | --- | --- | --- | --- | --- | --- | --- | --- | --- | --- | --- | --- |
| | | Math500 | Minerva | Olympia | AMC | Avg. | MMLU | MMLU-Pro | BBH | Avg. | ACPBench | HeadQA | Avg. |
| Qwen2.5-3B | Full | 60.80 | 26.10 | 23.26 | 35.00 | 36.29 | 64.13 | **38.66** | 52.29 | 51.69 | 32.68 | 62.69 | 47.69 |
| | GLOW | **62.80** | **27.21** | **24.30** | **42.50** | **39.20** | **64.49** | 38.63 | **53.20** | **52.11** | **33.66** | **63.38** | **48.52** |
| Qwen2.5-7B | Full | 76.60 | 40.07 | 38.96 | 55.00 | 52.66 | 72.24 | 53.71 | 70.84 | 65.60 | 38.27 | 72.06 | 55.17 |
| | GLOW | **79.60** | **40.07** | **41.04** | **60.00** | **55.18** | **73.99** | **55.77** | **71.99** | **67.25** | **39.19** | **72.50** | **55.85** |
| Qwen2.5-14B | Full | 86.80 | 47.79 | 52.30 | 82.50 | 67.35 | 81.56 | 67.63 | 80.90 | 76.70 | 48.13 | 81.44 | 64.79 |
| | GLOW | **87.80** | **52.21** | **52.44** | 82.50 | **68.74** | **82.53** | **68.70** | **81.65** | **77.63** | **49.51** | **82.35** | **65.93** |
| Qwen2.5-32B | Full | 92.20 | 52.57 | 57.19 | 85.00 | 71.74 | 85.22 | 73.10 | 83.53 | 80.62 | 50.67 | 84.90 | 67.79 |
| | GLOW | **93.40** | **54.41** | **59.11** | **92.50** | **74.86** | **85.51** | **74.14** | **83.98** | **81.21** | **51.97** | **85.19** | **68.58** |
| Llama3.1-8B | Full | 41.20 | 18.01 | 14.67 | 15.00 | 22.22 | 62.48 | 36.88 | 55.12 | 51.49 | 32.96 | 65.90 | 49.43 |
| | GLOW | **44.60** | **20.59** | **15.11** | **17.50** | **24.45** | **63.80** | **38.34** | **58.17** | **53.44** | **35.04** | **66.70** | **50.87** |

Table 7: Cross-domain performance of models trained on the **general reasoning** dataset. "Avg." denotes the average score within each group. **Bold** indicates the best results under the same model.

| Model | Setting | Math Reasoning (Out-of-Domain) | | | | | General Reasoning (In-Domain) | | | | Other Reasoning (Out-of-Domain) | | |
| --- | --- | --- | --- | --- | --- | --- | --- | --- | --- | --- | --- | --- | --- |
| | | Math500 | Minerva | Olympia | AMC | Avg. | MMLU | MMLU-Pro | BBH | Avg. | ACPBench | HeadQA | Avg. |
| Qwen2.5-3B | Full | 58.20 | 23.16 | 25.19 | 35.00 | 35.39 | 66.74 | 40.82 | **53.35** | 53.64 | 35.70 | 67.61 | 51.66 |
| | GLOW | **61.40** | **29.41** | **25.78** | **40.00** | **39.15** | **67.09** | **41.27** | 52.61 | **53.66** | **36.20** | **69.15** | **52.68** |
| Qwen2.5-7B | Full | 75.60 | 38.60 | 40.15 | 47.50 | 50.46 | 73.14 | **51.15** | 71.30 | 65.20 | 42.18 | 72.76 | 57.47 |
| | GLOW | **78.20** | **41.18** | **43.70** | **60.00** | **55.77** | **74.51** | 51.13 | **71.99** | **65.88** | **43.56** | **75.35** | **59.46** |
| Qwen2.5-14B | Full | 82.20 | 43.01 | 51.85 | 70.00 | 61.77 | 78.13 | 59.57 | 80.56 | 72.75 | 48.87 | 79.94 | 64.41 |
| | GLOW | **85.00** | **48.09** | **54.22** | 70.00 | **64.33** | **79.97** | **62.78** | **82.32** | **75.02** | **50.95** | **82.20** | **66.58** |
| Qwen2.5-32B | Full | 86.60 | 46.69 | 55.70 | 80.00 | 67.25 | 79.06 | 61.15 | 79.94 | 73.38 | 49.89 | 83.01 | 66.45 |
| | GLOW | **89.00** | **47.06** | **58.67** | **82.50** | **69.31** | **80.81** | **64.72** | **81.98** | **75.84** | **52.08** | **83.73** | **67.91** |
| Llama3.1-8B | Full | 20.00 | 15.81 | 6.52 | 2.50 | 11.21 | 66.49 | 40.56 | 53.73 | 53.59 | 36.06 | 69.55 | 52.81 |
| | GLOW | **24.80** | **20.59** | **6.96** | **12.50** | **16.21** | **68.52** | **42.96** | **57.53** | **56.33** | **39.72** | **72.57** | **56.15** |

**GLOW enhances cross-domain generalization without pre-selecting samples.** We apply GLOW to the random shuffled mixture of positive and negative data and observe consistent improvements across domains and different scales of models. For simplicity, we only report results for full and GLOW. For training results using standard SFT on positive-only and negative-only samples, please refer to Table 1 and Table 2. As shown in Table 6, GLOW surpasses standard SFT in-domain across all math-trained models and attains the best average on out-of-domain tasks. On Qwen2.5-7B it reaches 55.18 in-domain and 67.25 out-of-domain, while remaining competitive on general reasoning. Models trained on the general reasoning dataset also exhibit clear overall gains. Table 7 further reports that on Qwen2.5-14B, GLOW lifts out-of-domain math from 61.77 to 64.33 and out-of-domain reasoning from 64.41 to 66.58. These results indicate stronger data use from leveraging all samples and consistent improvements in both settings.

**GLOW typically assigns higher weights to negatives.** As shown in Figure 3, we train Qwen2.5-3B on math and MMLU tasks using GLOW. We use only questions and direct answers (for correctness checking) from these datasets, with all responses distilled from Qwen3-8B. The figure shows the fraction of negatives among examples receiving larger weights at each epoch. During Math and MMLU training, this fraction stays above 50% for most epochs, reaches about 75% to 80% early in training, and then decreases as learning progresses, but stays near 50%. This occurs because GLOW assigns larger weights to examples with stagnant loss reduction, a condition more common among negative samples. As a result, training places greater emphasis on unresolved reasoning rather than easy positives.

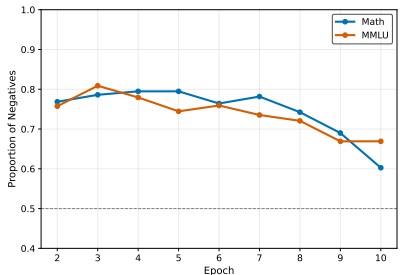

Figure 3: Fraction of negatives in the subset with the larger weights over epochs on Math and MMLU training.

**GLOW enhances reasoning exploration while maintaining answer decisiveness.** As shown in Table 8, applying GLOW consistently increases the average entropy during the thinking phase across all settings. For instance, think entropy rises from 0.36 to 0.71 on Math-to-Math and from 0.96 to 1.44 on MMLU-to-Other. In contrast, answer entropy changes modestly and even decreases out-of-domain. Taken together, these effects show that GLOW promotes broader exploration in reasoning while preserving answer decisiveness, which benefits generalization.

Table 8: Policy entropy changes with and without GLOW under various settings.

| Setting | Train | Test | $\bar{H}_{\text{think}}$ | $\bar{H}_{\text{ans}}$ | $\Delta H$ |
|---------|-------|------|------|------|------|
| Full | Math | Math | 0.36 | 0.22 | 0.14 |
| | | Other | 1.24 | 1.38 | -0.14 |
| | MMLU | Math | 0.54 | 0.34 | 0.20 |
| | | Other | 0.96 | 0.98 | -0.02 |
| GLOW | Math | Math | 0.71 | 0.35 | 0.36 |
| | | Other | 1.52 | 1.30 | 0.22 |
| | MMLU | Math | 0.89 | 0.52 | 0.37 |
| | | Other | 1.44 | 1.21 | 0.23 |

## 6 CONCLUSION

We show that negative reasoning trajectories can improve SFT generalization, mitigating the out-of-domain weakness of conventional training. Through analyses of data, training, and inference, we explain why negatives improve OOD generalization. Building on these insights, we introduce Gain-based LOss Weighting (GLOW), an adaptive, sample-aware scheme that up-weights underexplored examples by rescaling losses according to inter-epoch progress. Experiments demonstrate more data-efficient learning and consistent generalization gains across models and tasks.

## ETHICS STATEMENT

This work does not involve human subjects, sensitive personal data, or potentially harmful applications. The datasets used in our experiments are derived from publicly available resources and follow their respective licenses. We do not foresee ethical risks or violations associated with our methodology or findings.

## REPRODUCIBILITY STATEMENT

We provide detailed descriptions of our model selection, training objectives, and experimental setups in the main paper. Hyperparameters, dataset composition, and additional implementation details are included in the appendix. To further facilitate reproducibility, we will release our code through the URL referenced in the abstract.

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

# A APPENDIX

## A.1 EXPERIMENTS SETUP

**Distillation data curation** We conduct experiments on mathematical reasoning and common sense, using Qwen3-8B (Yang et al., 2025b) to distill reasoning trajectories. For mathematics, we collect data from OpenMathReasoning (Moshkov et al., 2025), and for common sense from MMLU (Hendrycks et al., 2021b;a). Each trajectory is labeled as positive if the final answer matches the ground truth and negative otherwise. To ensure that all samples preserve complete reasoning structures and differ only in correctness, we discard instances exceeding 8,192 tokens. We then sample positive and negative data in a 1:1 ratio, resulting in 7.2k instances for mathematics and 17.4k for common sense.

**Training Details** We conduct experiments on the Qwen2.5 series (3B, 7B, 14B, 32B) (Team, 2024) and LLaMA-3.1-8B(Dubey et al., 2024). All models are fine-tuned for 20 epochs with a batch size of 128, using a cosine learning rate scheduler with 10% warm-up steps and a maximum learning rate of $5 \times 10^{-5}$. We set the training length to 20 epochs, as the loss does not converge earlier and benchmark performance continues to improve up to this point.

**Evaluation Details** Following Huan et al. (2025); Yuan et al. (2025), we evaluate models on three categories of benchmarks: (1) **mathematical reasoning**: MATH500 (Hendrycks et al., 2024), OlympiaBench (He et al., 2024), MinervaMath (Lewkowycz et al., 2022), and the competition-level AMC2023 (Art of Problem Solving Foundation, 2023); (2) **common sense reasoning**: MMLU, MMLU-Pro (Wang et al., 2024b), and BBH (Suzgun et al., 2022); (3) **other OOD reasoning**: ACPBench (Kokel et al., 2025) for planning, and HeadQA (Vilares & Gómez-Rodríguez, 2019) for medicine. Model performance is measured by accuracy. Evaluation uses the codebase from (Yuan et al., 2025), with sampling temperature 0.6, top-p 0.95, one sample per input, and max generation length 32,768 tokens.

We define in-domain and out-of-domain (OOD) evaluation based on the training data distribution. For models fine-tuned on mathematical reasoning tasks, in-domain evaluation uses mathematical problems while OOD evaluation employs other task categories. Conversely, models trained on MMLU are evaluated in-domain on commonsense tasks and OOD on the remaining domains. We compare three training strategies: using only positive samples, only negative samples, and a balanced combination of both.

## A.2 DETAILED THEORETICAL DERIVATION

We provide a detailed derivation explaining why the dynamic reweighting mechanism in Eq. 6 improves optimization conditioning and, under standard assumptions, leads to improved generalization guarantees. The argument proceeds through a sequence of lemmas establishing: (i) the link between the loss-reduction statistic $\Delta_i^{(t)}$ and gradient alignment, (ii) the positive semi-definite (PSD) augmentation of the empirical Fisher induced by positive weight increments, (iii) a quantitative improvement of the spectrum of the Fisher in low-energy subspaces, and (iv) the transfer of improved conditioning to stability and generalization.

Throughout training, the underlying target objective remains the uniform empirical risk

$$R(\theta) = \frac{1}{N} \sum_{i=1}^{N} \ell_i(\theta).$$

However, the update direction at iteration $t$ is the gradient of the reweighted surrogate objective

$$R_w^{(t)}(\theta) = \frac{1}{N} \sum_{i=1}^{N} w_i^{(t)} \ell_i^{(t)}(\theta),$$

whose weights $w_i^{(t)}$ are dynamically adjusted by the reweighting rule. Accordingly, all conditioning and curvature statements in this section refer to the local quadratic model of $R_w^{(t)}$ at iteration $t$, rather than to the fixed uniform objective $R(\theta)$. Our results therefore characterize how the reweighting mechanism reshapes the second-order geometry of the surrogate objective used at each step.

### A.2.1 NOTATION AND STANDING ASSUMPTIONS

We keep the notation from the main text. At iteration $t$, we write $\theta^{(t)}$ for the current parameters, $g_i^{(t)} = \nabla_\theta \ell_i(\theta^{(t)})$ for the per-example gradients, and $w_i^{(t)}$ for the corresponding weights. For notational simplicity, we fix an iteration $t$ and often drop the superscript $(t)$ when it is clear from context. In particular, we write

$$g_i \triangleq g_i^{(t)} = \nabla_\theta \ell_i(\theta^{(t)}), \qquad G \triangleq G^{(t)} = \frac{1}{N} \sum_i w_i g_i.$$

We use $H_i(\theta) = \nabla_\theta^2 \ell_i(\theta)$ for the per-example Hessians, and we denote by

$$R_w(\theta) \triangleq \frac{1}{N} \sum_{i=1}^N w_i \, \ell_i(\theta)$$

the reweighted surrogate objective at iteration $t$ (with weights $\{w_i\}$ held fixed). Its Hessian is

$$H(\theta) \triangleq \nabla_\theta^2 R_w(\theta).$$

The empirical (weighted) Fisher at the same iteration is

$$F_w(\theta) \triangleq \frac{1}{N} \sum_{i=1}^N w_i \, g_i g_i^\top,$$

and, since we work at a fixed iteration, we often abbreviate $F_w(\theta^{(t)})$ to $F_w$.

We now collect the assumptions used in the analysis.

**Assumption A.1** (Smoothness, boundedness, curvature, and energy injection).

(A1) *Each $\ell_i(\theta)$ is twice differentiable and $L$-smooth: $\|H_i(\theta)\|_{\mathrm{op}} \le L$.*

(A2) *Gradient norms are uniformly bounded: $\|g_i(\theta)\|_2 \le G_{\max}$.*

(A3) *The learning rate $\eta$ is small enough that higher-order terms are controlled.*

(A4) *(Fisher–Hessian closeness for $R_w$) In the local region of interest and for all iterates $\theta$ visited by the algorithm, the Hessian $H(\theta) = \nabla_\theta^2 R_w(\theta)$ of the reweighted objective and the corresponding empirical Fisher $F_w(\theta)$ satisfy*
$$\|H(\theta) - F_w(\theta)\|_{\mathrm{op}} \le \delta.$$

(A5) *(Energy injection of the reweighting rule) Let $U$ be a $k$-dimensional low-curvature subspace with orthogonal projector $P_U$. At each step, let $T$ denote the set of examples whose weights are increased, with nonnegative increments $\delta w_i \ge 0$ for $i \in T$. The induced change in the empirical Fisher is*
$$\Delta F \triangleq \frac{1}{N} \sum_{i \in T} \delta w_i \, g_i g_i^\top.$$

*We assume that the reweighting rule injects curvature uniformly into $U$ in the sense that for every unit vector $v \in U$,*
$$v^\top \Delta F v \ge \frac{\gamma}{k}.$$
*Equivalently, the restriction of $\Delta F$ to $U$ satisfies $\Delta F|_U \succeq (\gamma/k) P_U$, and in particular*
$$\mathrm{tr}(P_U \Delta F P_U) = \mathrm{tr}(\Delta F|_U) \ge \gamma.$$

**Lemma A.1** (Taylor relation between $\Delta_i$ and gradient alignment). *Under assumptions (A1)–(A3), after one update $\theta \leftarrow \theta - \eta G$,*

$$\Delta_i \;=\; \ell_i(\theta) - \ell_i(\theta - \eta G) = \eta \, g_i^\top G - \frac{1}{2} \eta^2 \, G^\top H_i(\xi_i) G,$$

*for some $\xi_i$ on the line segment between $\theta$ and $\theta - \eta G$. Moreover,*

$$\left| \Delta_i - \eta \, g_i^\top G \right| \le \frac{1}{2} L \eta^2 \|G\|_2^2.$$

*Proof.* Second-order Taylor expansion yields the stated form, and $L$-smoothness gives the remainder bound. $\qquad\square$

**Lemma A.2** (Positive weight increments induce PSD augmentation). *Let the weights change by nonnegative increments $\delta w_i \geq 0$ for $i \in T$. The induced change in the empirical Fisher is*

$$\Delta F = \frac{1}{N} \sum_{i \in T} \delta w_i \, g_i g_i^\top,$$

*which is PSD. Consequently, the updated Fisher $F_w' = F_w + \Delta F$ satisfies $F_w' \succeq F_w$, and, when the eigenvalues of both matrices are ordered in nondecreasing order, we have $\lambda_j(F_w') \geq \lambda_j(F_w)$ for all $j$.*

*Proof.* Each outer product $g_i g_i^\top$ is symmetric and PSD. Since $\delta w_i \geq 0$, every term $\delta w_i g_i g_i^\top$ is PSD, and their average $\Delta F$ is PSD as well. Thus $F_w' = F_w + \Delta F$ is a PSD perturbation of the symmetric matrix $F_w$, so by Weyl's eigenvalue inequality we obtain $\lambda_j(F_w') \geq \lambda_j(F_w)$ for all $j$ when the eigenvalues are ordered in nondecreasing order. $\qquad\square$

To evaluate how reweighting affects curvature in directions where the objective is weakly curved, we consider a $k$-dimensional subspace $U$ spanned by small-eigenvalue directions of the empirical Fisher $F_w$. Introducing such a subspace is standard in conditioning analysis, as the restricted spectrum $F_w|_U$ precisely characterizes curvature along these low-eigenvalue directions. Let $P_U$ denote the orthogonal projector onto $U$. Intuitively, directions associated with large eigenvalues of $F_w$ already exhibit sufficient curvature and are repeatedly explored by gradient-based updates. In contrast, the low-eigenvalue subspace $U$ captures flat or poorly conditioned directions that act as the main bottleneck for optimization and conditioning. Our analysis therefore focuses on how reweighting increases curvature within $U$ rather than on further amplifying already well-conditioned directions.

The effect of reweighting on second-order geometry is captured entirely by the increment

$$\Delta F = \frac{1}{N} \sum_{i \in T} \delta w_i \, g_i g_i^\top,$$

which is positive semi-definite by construction: each $g_i g_i^\top$ is PSD and each weight increment $\delta w_i$ arising from the reweighting rule equation 6 is nonnegative. Hence the updated Fisher satisfies $F_w' = F_w + \Delta F \succeq F_w$, providing a monotone PSD augmentation that allows the application of standard Weyl-type eigenvalue inequalities.

To guarantee that this PSD increment has a meaningful effect on the low-curvature subspace $U$, we impose the uniform energy condition in (A5): there exists a constant $\gamma > 0$ such that, for every unit vector $v \in U$ (where $U$ is $k$-dimensional with orthogonal projector $P_U$),

$$v^\top \Delta F v \;\geq\; \frac{\gamma}{k}.$$

Equivalently, the restriction $\Delta F|_U$ satisfies $\Delta F|_U \;\succeq\; (\gamma/k)\, P_U$, so that $\operatorname{tr}(P_U \Delta F P_U) = \operatorname{tr}(\Delta F|_U) \geq \gamma$ as a simple corollary. Intuitively, this condition rules out the degenerate case in which all of the additional mass is concentrated on a few directions inside $U$; instead, it enforces a uniform strengthening of curvature across the low-eigenvalue subspace. We now state the resulting spectral improvement.

**Lemma A.3** (Improvement of small-eigenvalue subspace). *Let $U$ be a $k$-dimensional subspace with projector $P_U$. Suppose the weight increments satisfy the energy condition (A5), so that $\Delta F|_U \succeq (\gamma/k)\, P_U$ for some $\gamma > 0$. Let $\lambda_{\min}(F_w|_U)$ denote the minimal eigenvalue of $F_w$ restricted to $U$. Then the minimal eigenvalue satisfies*

$$\lambda_{\min}(F_w'|_U) \geq \lambda_{\min}(F_w|_U) + \frac{\gamma}{k}.$$

*Proof.* By (A5) we have $\Delta F|_U \succeq (\gamma/k)\, P_U$, which implies $\lambda_{\min}(\Delta F|_U) \geq \gamma/k$. Since $F_w'|_U = F_w|_U + \Delta F|_U$ and both are symmetric, the eigenvalue monotonicity for sums of Hermitian matrices yields

$$\lambda_{\min}(F_w'|_U) \;\geq\; \lambda_{\min}(F_w|_U) + \lambda_{\min}(\Delta F|_U) \;\geq\; \lambda_{\min}(F_w|_U) + \frac{\gamma}{k},$$

as claimed. $\qquad\square$

*Remark* A.4. The uniform energy condition in (A5) ensures that the augmented weights inject non-trivial curvature in every direction of $U$, not just along a few isolated eigenvectors. This rules out pathological cases where the trace increases but the smallest eigenvalue remains nearly unchanged, and guarantees a genuine improvement of the worst-case curvature on $U$.

**Lemma A.5** (Transfer from Fisher to Hessian). *Under (A4), if*

$$\lambda_{\min}(F'_w|_U) - \lambda_{\min}(F_w|_U) \geq \Delta\lambda_F,$$

*then the local Hessian of the reweighted objective satisfies*

$$\lambda_{\min}(H'|_U) \geq \lambda_{\min}(H|_U) + \Delta\lambda_F - 2\delta,$$

*where $H$ and $H'$ denote the Hessian $H(\theta) = \nabla_\theta^2 R_w(\theta)$ evaluated at the current and updated parameters, respectively.*

*Proof.* Assumption (A4) yields $\|H - F_w\|_{\mathrm{op}} \leq \delta$ and $\|H' - F'_w\|_{\mathrm{op}} \leq \delta$ at the current and updated iterates. These bounds imply matching eigenvalue relations before and after the update, giving the stated inequality. □

**Lemma A.6** (Improved conditioning reduces parameter sensitivity). *Assume a restricted strong convexity condition on $U$: there exists $\mu > 0$ such that, throughout the local region,*

$$\lambda_{\min}(H|_U) \geq \mu.$$

*Consider two training sets that differ by a single example and run identical reweighted updates. Under standard Lipschitz assumptions on the gradients, the resulting parameter perturbation between the two runs is $O(1/\mu)$. Hence increasing $\mu$—equivalently improving the smallest eigenvalue of $H$ on $U$—reduces algorithmic instability and yields a smaller generalization gap.*

*Proof sketch.* Restricted strong convexity with parameter $\mu$ implies that the reweighted surrogate objective $R_w$ is $\mu$-strongly convex along directions in $U$. In particular, the map that sends the empirical risk (or its gradient) to its minimizer is $1/\mu$-Lipschitz along $U$: if two datasets differ by one example, the corresponding empirical gradients differ by at most a constant $L_g$, and the resulting parameters $\theta$ and $\theta'$ satisfy

$$\|\theta' - \theta\| \leq \frac{L_g}{\mu}.$$

This $O(1/\mu)$ sensitivity of the iterates yields uniform stability, in the sense that the loss on any test point differs by at most $O(1/\mu)$ between the two runs. The uniform stability framework of Bousquet & Elisseeff (2002) and the refinement in Hardt et al. (2016) then imply that such a stability bound translates into an $O(1/\mu)$ upper bound on the generalization error. We refer to these works for complete statements and proofs. □

**Proposition A.7** (Main result: conditioning and generalization improvement). *Under (A1)–(A5), suppose the weight update rule equation 6 produces nonnegative increments satisfying the energy condition (A5). Then:*

1. *The empirical Fisher receives a PSD augmentation $\Delta F$ and, on the low-curvature subspace $U$, both its average eigenvalue and its minimal eigenvalue increase by at least $\gamma/k$:*

$$\frac{1}{k}\operatorname{tr}(P_U F'_w P_U) \geq \frac{1}{k}\operatorname{tr}(P_U F_w P_U) + \frac{\gamma}{k}, \qquad \lambda_{\min}(F'_w|_U) \geq \lambda_{\min}(F_w|_U) + \frac{\gamma}{k}.$$

   *The first inequality follows from the trace identity and (A5), while the second is the content of Lemma A.3.*

2. *By Lemma A.5, taking $\Delta\lambda_F = \gamma/k$ as provided by Lemma A.3, the minimal Hessian eigenvalue of the reweighted objective on $U$ satisfies*

$$\lambda_{\min}(H'|_U) \geq \lambda_{\min}(H|_U) + \frac{\gamma}{k} - 2\delta.$$

3. *If the local loss satisfies a restricted strong-convexity condition on U with parameter μ, i.e.* $\lambda_{\min}(H|_U) \geq \mu$, *then after reweighting we can take*

$$\mu' \triangleq \mu + \frac{\gamma}{k} - 2\delta,$$

*which improves algorithmic stability and yields a sharper stability-based generalization bound according to Lemma A.6.*

*Proof.* The conclusions follow by combining Lemmas A.2, A.3, A.5, and A.6. □

In summary, the dynamic reweighting mechanism amplifies underrepresented gradients and induces a PSD augmentation of the empirical Fisher, thereby improving the conditioning of the local quadratic model of the reweighted objective $R_w^{(t)}$ at each iteration. Through the stability analysis of Lemma A.6, this improved per-iteration conditioning translates into more stable optimization dynamics and sharper stability-based generalization guarantees.

### A.3 DETAILED TAXONOMY OF NEGATIVE TRAINING SAMPLES

We provide statistics on the detailed categorization of negative samples in our training dataset. As shown in Figure 4a and Figure 4b, the error types of samples from OpenMathReasoning and MMLU that are not selected by reject sampling can be grouped into nine major categories and twenty-four subcategories. Although the distribution across categories is imbalanced, the errors still exhibit a broad coverage, ensuring a comprehensive representation of error types.

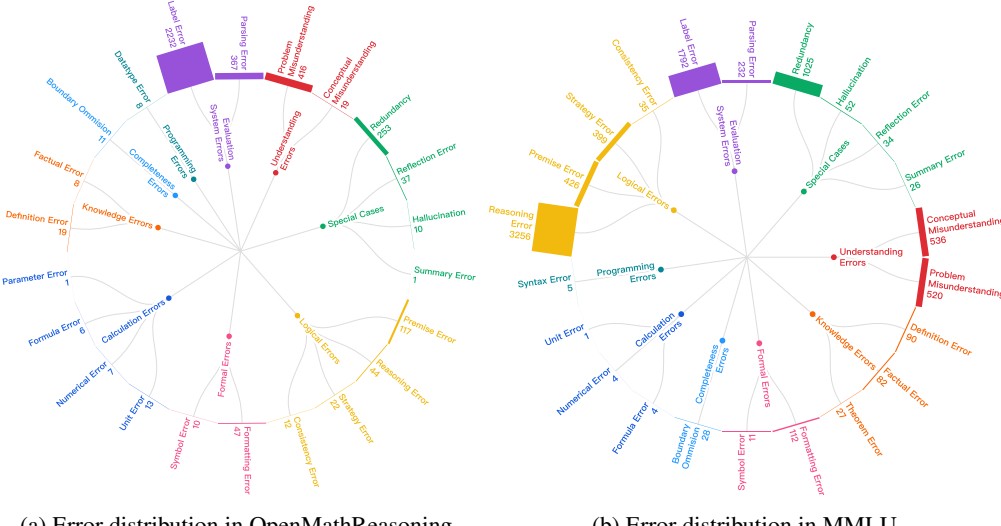

(a) Error distribution in OpenMathReasoning.  (b) Error distribution in MMLU.

Figure 4: Detailed categorization of negative samples in OpenMathReasoning and MMLU.

### A.4 HYPERPARAMETER SENSITIVITY OF GLOW

As shown in Figure 5, GLOW yields modest improvements over the full-SFT reference in most configurations. Varying $\alpha$ between 0.8 and 1.5 leads to small changes, and $\beta = 12$ is generally stronger than $\beta = 10$ or $\beta = 18$ at matched $\alpha$. These results suggest incremental gains with moderate hyperparameter choices in our setup.

### A.5 PROMPT FOR CATEGORIZE NEGATIVE SAMPLES

We design a structured prompt to categorize each erroneous reasoning trajectory into a fine-grained error class. The classification framework contains 9 primary categories and 22 sub-categories. The full classification schema and the prompt used for categorization are shown in Figure 6.

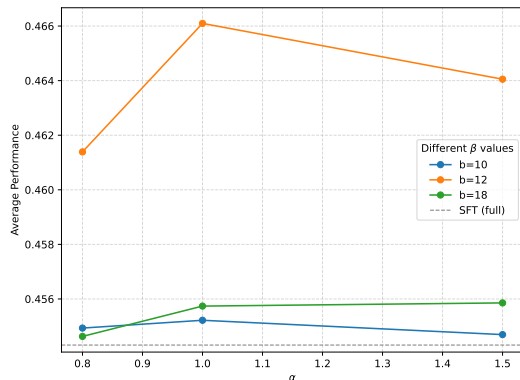

Figure 5: Ablation study on the hyperparameters $\alpha$ and $\beta$. GLOW exhibits stable performance across different settings, demonstrating the robustness of the reweighting formulation.

### A.6 CASE STUDY

As discussed in Section 4.3, negative trajectories exhibit higher entropy than positives ones on certain reasoning tokens and transition words. For illustration, we select one case and highlight the high-entropy segments. The results show that negatives contain substantially more such reasoning-related high-entropy fragments than positives.

### A.7 CASE STUDY OF NEGATIVE SAMPLES

As discussed in Section 4.3, negative trajectories exhibit higher entropy than positives ones on certain reasoning tokens and transition words. For illustration, we select one case and highlight the high-entropy segments. The results show that negatives contain substantially more such reasoning-related high-entropy fragments than positives.

### A.8 TRAINING LOSS ON OPENMATHREASONING AND MMLU

We present in Figure 9 the loss comparison of all models trained under the positive and negative settings on the OpenMathReasoning and MMLU datasets.

### A.9 PROGRESS LOSS

Table 9 reports intermediate checkpoint evaluations for Qwen2.5-7B and Qwen2.5-32B trained on the math reasoning and general reasoning datasets. For each setting, we compare SFT using positive distilled reasoning trajectories against SFT using negative distilled reasoning trajectories at 5, 10, 15, and 20 epochs. Across model sizes and training sets, negative-trajectory SFT consistently improves over the base model and shows gains that are comparable to the positive-trajectory counterpart. In several configurations, negative-trajectory SFT matches or exceeds positive-trajectory SFT on out-of-domain benchmarks. These results suggest that negative trajectories contain structured supervision signal rather than noise.

### A.10 THE USE OF LARGE LANGUAGE MODELS (LLMS)

We used LLMs only for copy-editing and minor stylistic polishing (grammar, phrasing, and LaTeX formatting). All suggestions were manually reviewed and edited by the authors. The authors take full responsibility for the manuscript's contents.

---

**Prompt for Categorizing Negative Samples**

```
You are an expert AI assistant tasked with identifying the single,
most specific error category from the list below.

Error Category List:
- Primary_category: Understanding Errors
  - sub_category: Problem Misunderstanding, Conceptual
      Misunderstanding
- Primary_category: Knowledge Errors
  - sub_category: Factual Error, Theorem Error, Definition Error
- Primary_category: Logical Errors
  - sub_category: Strategy Error, Reasoning Error, Premise Error,
      Consistency Error
- Primary_category: Calculation Errors
  - sub_category: Numerical Error, Formula Error, Parameter Error,
      Unit Error
- Primary_category: Programming Errors
  - sub_category: Syntax Error, Function Error, Data Type Error
- Primary_category: Formal Errors
  - sub_category: Symbol Error, Formatting Error
- Primary_category: Completeness Errors
  - sub_category: Boundary Omission
- Primary_category: Special Cases
  - sub_category: Reflection Error, Summary Error, Hallucination,
      Redundancy
- Primary_category: Evaluation System Errors
  - sub_category: Incorrect Ground Truth, Correct Answer Parsing
      Error

Data for Analysis:
- Question: {question}
- Ground Truth Answer: {groundtruth}
- Model's Reasoning Process (to be analyzed): {model_reasoning}

CRITICAL INSTRUCTION:
Analyze the provided reasoning process. Your response MUST be ONLY
    a single,
raw JSON object with the keys "sub_category" and "analysis". Do not
    include any
other text, explanations, apologies, or markdown formatting.

Example of a perfect response:
{
  "sub_category": "Premise Error",
  "analysis": "The model incorrectly assumed that all bicycles use
      plastic
            squares for identification, which is a flawed premise
                not
            supported by the question's context."
}
```

Figure 6: Prompt used for categorizing negative reasoning samples into predefined error subcategories.

**An Example from MATH-500**

**Problem**: Denali and Nate work for a dog walking business and are paid for each dog they walk. Denali is responsible for 16 dogs and Nate is responsible for 12 dogs. Under the company's new policy, they will be assigned or unassigned new dogs in groups of $x$ dogs. The ratio of Denali's pay to Nate's pay would be the same if Denali started walking $4x$ more dogs and Nate stayed at 12 dogs or if $x$ of Nate's dogs were reassigned to Denali. Find $x$ if $x \neq 0$. **Answer**: $\boxed{5}$

| $M_{pos}$ **Thinking Trajectories** | $M_{neg}$ **Thinking Trajectories** |
|---|---|
| Okay, let me try to figure out this problem... 
 So, setting up the equation... 
 Let me compute both sides... 

 ...would be the same under two different scenarios. Let me parse that... 

 So, solution is x = 0... 

 But the problem says ... 

 Let me check again... 

 x = 0 is correct... 
 The final answer is $\boxed{0}$. | Okay, let me try to figure out this problem... 
 The key is that... 
 Now, we can solve this... 
 ..., so the answer is x = 3. 
 Wait, let me check... 
 it is wrong... 

 *Alternatively, maybe there's a different way to interpret the problem?...* 
 Maybe I missed something... 

 *Alternatively, maybe there's a different interpretation? Let me check...* 
 Wait, no. The problem says... 
 ..., so the answer is 5. 
 But, Let me check ... 
 Therefore, x = 5 is correct... 
 The final answer is $\boxed{5}$. |

Figure 7: Case study of thinking trajectories for $M_{pos}$ and $M_{neg}$ on the same question.

**Problem**

The graph shows the total distance Sam drove from 6 a.m. to 11 a.m. How many miles per hour is the car's average speed for the period from 6 a.m. to 11 a.m.?

unitsize(0.2inch); draw((0,0)–(5.5,0)); draw((0,0)–(0,8.5)); draw((1,0)–(1,8.5)); draw((2,0)–(2,8.5)); draw((3,0)–(3,8.5)); draw((4,0)–(4,8.5)); draw((5,0)–(5,8.5)); draw((0,1)–(5.5,1)); draw((0,8)–(5.5,8)); draw((0,7)–(5.5,7)); draw((0,6)–(5.5,6)); draw((0,5)–(5.5,5)); draw((0,4)–(5.5,4)); draw((0,3)–(5.5,3)); draw((0,2)–(5.5,2)); draw((0,0)–(1,2)–(2,3)–(3,5)–(4,6)–(5,8)); dot((0,0)); dot((1,2)); dot((2,3)); dot((3,5)); dot((4,6)); dot((5,8)); label("6",(0,-0.5),S); label("7",(1,-0.5),S); label("8",(2,-0.5),S); label("9",(3,-0.5),S); label("10",(4,-0.5),S); label("11",(5,-0.5),S); label("0",(-0.5,0),W); label("40",(-0.5,2),W); label("80",(-0.5,4),W); label("120",(-0.5,6),W); label("160",(-0.5,8),W); label("Time of Day (a.m.)",(2.7,-2),S); label("Total distance",(-0.5,9),N);

**Model's Thinking Trajectory and Answer**

Let me start with the total time. The period is from 6 a.m. to 11 a.m. Let me count the hours. From 6 to 7 is 1 hour, 7 to 8 is another, 8 to 9, 9 to 10, and 10 to 11. That's 5 hours total. Wait, is that right? Let me check again. From 6 to 11 is 5 hours? Wait, 11 minus 6 is 5, yes. So total time is 5 hours. Okay, that seems straightforward.

Now, the total distance. The graph shows the total distance driven at different times. The Asymptote code draws a graph with time on the x-axis and total distance on the y-axis. The points given are at time 6 a.m. (which is the first point, x = 0), then at 7 a.m. (x = 1), 8 a.m. (x = 2), etc., up to 11 a.m. (x = 5). The corresponding distances are given as y-values. Let me check the Asymptote code again to see the coordinates.

Looking at the Asymptote code, the graph is drawn from $(0, 0)$ to $(1, 2)$ to $(2, 3)$ to $(3, 5)$ to $(4, 6)$ to $(5, 8)$. The labels on the y-axis are 0, 40, 80, 120, 160. So each unit on the y-axis corresponds to 40 miles? Let me confirm. The labels are at $y = 0, 2, 4, 6, 8$. The labels next to them are 0, 40, 80, 120, 160. So, each vertical unit is 40 miles. Therefore, the y-coordinate multiplied by 40 gives the total distance in miles.

So, for example:

- At 6 a.m. (x = 0), the total distance is 0 miles.

- At 7 a.m. (x = 1), the total distance is 2 units, which is $2 \times 40 = 80$ miles.

- At 8 a.m. (x = 2), it's 3 units, so $3 \times 40 = 120$ miles.

- At 9 a.m. (x = 3), 5 units, which is $5 \times 40 = 200$ miles.

- At 10 a.m. (x = 4), 6 units, which is $6 \times 40 = 240$ miles.

- At 11 a.m. (x = 5), 8 units, which is $8 \times 40 = 320$ miles.

The question is about the total distance from 6 a.m. to 11 a.m. The graph is showing cumulative distance. So at 11 a.m., the total distance is 320 miles. Therefore, the total distance driven is 320 miles.

If the graph is cumulative, then the total distance at 11 a.m. is the total distance driven from 6 a.m. to 11 a.m., so that would be 320 miles. Therefore, total distance is 320 miles, total time is 5 hours. Therefore, average speed is $320 \div 5 = 64$ miles per hour.

Let me check if the total distance is indeed 320. The last point is at 11 a.m., which is $x = 5, y = 8$. Since each $y$ unit is 40 miles, $8 \times 40 = 320$. Yes. So that seems correct.

Alternatively, maybe the question is trying to trick me? Suppose someone might think that the total distance is the sum of all the segments, but because the graph is cumulative, the total distance at 11 a.m. is already the total distance driven from 6 a.m. to 11 a.m.

We can also confirm by looking at increments: From 6–7: 80 miles, From 7–8: $120 - 80 = 40$ miles. From 8–9: $200 - 120 = 80$ miles. From 9–10: $240 - 200 = 40$ miles. From 10–11: $320 - 240 = 80$ miles Summing gives $80 + 40 + 80 + 40 + 80 = 320$ miles, consistent with the final value.

So average speed is total distance divided by total time. Total time is 5 hours. $320 \div 5 = 64$, so 64 miles per hour.

The time from 6 a.m. to 11 a.m. is indeed 5 hours. The x-axis is labeled from 6 a.m. to 11 a.m., with each unit representing one hour, so that checks out.

Therefore, I think my answer is correct.

**Final Answer**: the car's average speed is $\boxed{64}$ miles per hour.

Figure 8: Case study of a negative trajectory from the OpenMathReasoning training dataset. The model misinterprets the problem, but its subsequent step-by-step reasoning and formula derivations remain structurally correct.

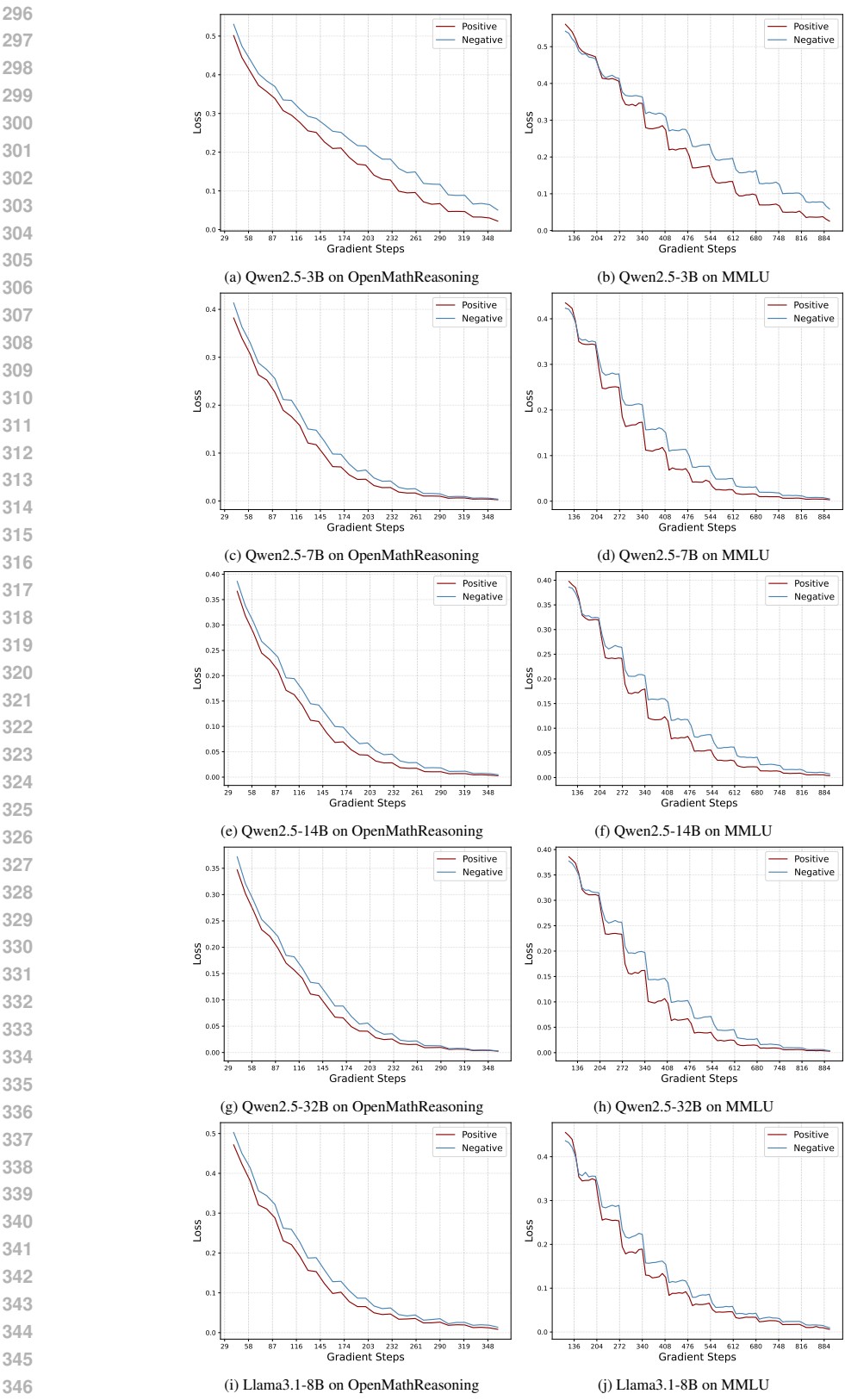

Figure 9: Training loss of Qwen2.5 models and Llama3.1-8B on OpenMathReasoning (left) and MMLU (right). Losses drop across epochs, with the positive setting converging faster than the negative.

Table 9: **Checkpoint evaluation across SFT epochs with distilled reasoning trajectories.** We report performance at 5, 10, 15, and 20 epochs. Each row corresponds to a model size and training dataset, and each row contains two subtables that compare training on positive (left) versus negative (right) distilled trajectories. Columns in each subtable correspond to benchmarks, while rows correspond to training epochs, with Base denoting the model before SFT.

(a) Qwen2.5-7B is fine-tuned on the **math reasoning** dataset using **positive** distilled trajectories.

| Epoch | Math500 | Minerva | Olympia | AMC | MMLU | MMLU-Pro | BBH |
|---|---|---|---|---|---|---|---|
| Base | 58.40 | 26.84 | 26.07 | 52.50 | 55.80 | 26.56 | 51.10 |
| 5epoch | 72.80 | 37.13 | 37.19 | 45.00 | 60.95 | 30.34 | 54.69 |
| 10epoch | 75.80 | 38.24 | 40.59 | 65.00 | 64.06 | 32.50 | 61.62 |
| 15epoch | 77.20 | 36.76 | 41.93 | 55.00 | 60.81 | 32.15 | 59.69 |
| 20epoch | 78.00 | 36.76 | 41.78 | 57.50 | 61.03 | 32.70 | 60.58 |

(b) Qwen2.5-7B is fine-tuned on the **math reasoning** dataset using **negative** distilled trajectories.

| Epoch | Math500 | Minerva | Olympia | AMC | MMLU | MMLU-Pro | BBH |
|---|---|---|---|---|---|---|---|
| Base | 58.40 | 26.84 | 26.07 | 52.50 | 55.80 | 26.56 | 51.10 |
| 5epoch | 71.20 | 31.99 | 31.56 | 47.50 | 62.58 | 44.04 | 56.28 |
| 10epoch | 77.20 | 34.93 | 39.26 | 50.00 | 71.39 | 52.14 | 69.49 |
| 15epoch | 78.60 | 39.71 | 38.37 | 52.50 | 72.10 | 52.24 | 71.09 |
| 20epoch | 77.60 | 40.44 | 38.37 | 57.50 | 73.11 | 53.74 | 71.73 |

(c) Qwen2.5-7B is fine-tuned on the **general reasoning** dataset using **positive** distilled trajectories.

| Epoch | Math500 | Minerva | Olympia | AMC | MMLU | MMLU-Pro | BBH |
|---|---|---|---|---|---|---|---|
| Base | 58.40 | 26.84 | 26.07 | 52.50 | 55.80 | 26.56 | 51.10 |
| 5epoch | 72.00 | 36.76 | 37.33 | 47.50 | 73.62 | 50.61 | 64.05 |
| 10epoch | 74.60 | 37.50 | 41.48 | 55.00 | 73.79 | 53.32 | 69.73 |
| 15epoch | 72.00 | 37.50 | 39.26 | 50.00 | 74.11 | 53.91 | 68.34 |
| 20epoch | 74.40 | 37.50 | 39.85 | 50.00 | 73.42 | 53.22 | 68.23 |

(d) Qwen2.5-7B is fine-tuned on the **general reasoning** dataset using **negative** distilled trajectories.

| Epoch | Math500 | Minerva | Olympia | AMC | MMLU | MMLU-Pro | BBH |
|---|---|---|---|---|---|---|---|
| Base | 58.40 | 26.84 | 26.07 | 52.50 | 55.80 | 26.56 | 51.10 |
| 5epoch | 76.80 | 36.76 | 37.78 | 47.50 | 71.09 | 43.99 | 66.00 |
| 10epoch | 76.80 | 37.87 | 40.30 | 52.50 | 71.43 | 45.87 | 68.84 |
| 15epoch | 76.80 | 37.13 | 41.48 | 55.00 | 71.30 | 44.62 | 69.30 |
| 20epoch | 77.00 | 37.13 | 42.07 | 60.00 | 71.23 | 45.79 | 69.46 |

(e) Qwen2.5-32B is fine-tuned on the **math reasoning** dataset using **positive** distilled trajectories.

| Epoch | Math500 | Minerva | Olympia | AMC | MMLU | MMLU-Pro | BBH |
|---|---|---|---|---|---|---|---|
| Base | 63.20 | 34.19 | 26.52 | 35.00 | 68.34 | 39.80 | 58.65 |
| 5epoch | 90.20 | 49.63 | 59.11 | 85.00 | 76.53 | 46.77 | 78.04 |
| 10epoch | 92.60 | 50.00 | 60.44 | 85.00 | 78.63 | 51.67 | 79.01 |
| 15epoch | 93.00 | 48.53 | 62.07 | 90.00 | 78.72 | 51.99 | 80.57 |
| 20epoch | 91.40 | 50.74 | 60.89 | 85.00 | 79.01 | 54.31 | 80.61 |

(f) Qwen2.5-32B is fine-tuned on the **math reasoning** dataset using **negative** distilled trajectories.

| Epoch | Math500 | Minerva | Olympia | AMC | MMLU | MMLU-Pro | BBH |
|---|---|---|---|---|---|---|---|
| Base | 63.20 | 34.19 | 26.52 | 35.00 | 68.34 | 39.80 | 58.65 |
| 5epoch | 88.40 | 45.22 | 52.30 | 85.00 | 83.07 | 68.23 | 83.55 |
| 10epoch | 92.20 | 51.10 | 57.93 | 85.00 | 85.14 | 73.75 | 84.22 |
| 15epoch | 91.20 | 50.74 | 57.33 | 90.00 | 85.02 | 73.48 | 84.62 |
| 20epoch | 92.20 | 50.74 | 58.37 | 95.00 | 85.47 | 73.53 | 84.51 |

(g) Qwen2.5-32B is fine-tuned on the **general reasoning** dataset using **positive** distilled trajectories.

| Epoch | Math500 | Minerva | Olympia | AMC | MMLU | MMLU-Pro | BBH |
|---|---|---|---|---|---|---|---|
| Base | 63.20 | 34.19 | 26.52 | 35.00 | 68.34 | 39.80 | 58.65 |
| 5epoch | 84.60 | 44.85 | 52.00 | 62.50 | 82.10 | 66.54 | 80.03 |
| 10epoch | 86.60 | 46.69 | 55.70 | 75.00 | 81.14 | 67.01 | 80.69 |
| 15epoch | 85.00 | 47.06 | 56.59 | 75.00 | 81.73 | 68.33 | 81.73 |
| 20epoch | 85.20 | 46.69 | 56.15 | 75.00 | 81.97 | 68.54 | 81.60 |

(h) Qwen2.5-32B is fine-tuned on the **math reasoning** dataset using **negative** distilled trajectories.

| Epoch | Math500 | Minerva | Olympia | AMC | MMLU | MMLU-Pro | BBH |
|---|---|---|---|---|---|---|---|
| Base | 63.20 | 34.19 | 26.52 | 35.00 | 68.34 | 39.80 | 58.65 |
| 5epoch | 85.00 | 44.49 | 51.26 | 77.50 | 78.74 | 57.48 | 79.09 |
| 10epoch | 87.20 | 46.30 | 54.52 | 75.00 | 79.01 | 60.43 | 80.88 |
| 15epoch | 86.40 | 47.79 | 55.70 | 65.00 | 77.77 | 57.14 | 79.97 |
| 20epoch | 86.40 | 47.06 | 56.89 | 72.50 | 77.99 | 58.34 | 80.71 |

## A.11 LIMITATION

Our study primarily examines gain-based reweighting in the supervised fine-tuning stage of reasoning post-training, and we leave its interaction with subsequent RLHF or other reinforcement learning stages as an exciting direction for future work. In addition, our experiments focus on text-only chain-of-thought data for math and multi-task knowledge benchmarks with a small set of open-source backbones, so a natural next step is to extend the same analysis and method to broader task families, larger model scales and multimodal or tool-augmented settings, building on the phenomena and gains established in this work.

