# OpenReview forum: "Learning from Mistakes: Negative Reasoning Samples Enhance Out-of-Domain Generalization"
_ICLR.cc/2026/Conference — ICLR 2026 Conference Withdrawn Submission_

### Official Review · Reviewer_SmfD · 2025-10-23

**Soundness:** 1
**Presentation:** 1
**Contribution:** 2
**Rating:** 2
**Confidence:** 3

**Summary:**

This paper investigates the practice of Supervised Fine-Tuning (SFT) of using only "positive" Chain-of-Thought (CoT) samples during distillation. The authors argue this approach leads to overfitting, poor out-of-domain (OOD) generalization, and inefficient data use by discarding "negative" (incorrect) samples. While the authors findings that OOD benchmarks perform better with negative samples is interesting, the paper requires major revisions. The communication is very challenging to follow; with some statements being incorrect. The problem setting’s impact is unclear, and experimental choices need clarification. Some analysis interpretation’s validity are also questionable.

**Strengths:**

1. The empirical finding is interesting.
2. The empirical analysis is interesting.

**Weaknesses:**

As the paper mentions, the pre-training + post-training (SFT, RL) is the common paradigm. There are clear evidence that shows that RL generalizes and SFT memorizes,e.g.
“SFT Memorizes, RL Generalizes: A Comparative Study of Foundation Model Post-training”
Modern post-training of reasoning models never stop at SFT, but always include the RL stage as it dramatically improves performance. Given that it is possible that RL improves generalization, for this work to be practically impactful it would need to show that the generalization gains at SFT lead to improved generalization or in-distribution performance post-RL.

First line of the abstract is confusing. The phrasing makes it sound like SFT always uses CoT, which is incorrect. SFT with CoT is only for reasoning post-training.

Why is there + in the data annotation in Fig. 1? This seems unnecessary as it is not increasing; it is just the performance annotation.

Typos should be fixed. Also, there should be a space before parenthesis, E.g.
> Chain-of-Thougnt(CoT)

> tasks(see

> benchmarks..

The communication is confusing. What does dominant SFT mean? Why is the idea of distillation suddenly introduced here?

> (2) Data Inefficiency: Dominant SFT relies on distilled reasoning paths and uses rejection sampling to select those leading to correct answers and formats

The text size in figures and diagrams should be increased in size. It is non-legible when printed out in A4 paper.

This phrase is confusing. The communication needs revision.
> While relying solely on negative samples is a possible strategy, it remains a form of rejection sampling with low data efficiency

It is not correct to call MMLU and MMLU-pro a common sense benchmark
> We use Qwen3-8B to distill responses from OpenMathReasoning (Moshkov et al., 2025) and the MMLU (Hendrycks et al., 2021b) training set as training data for mathematical and common sense tasks.


MMLU includes mathematics so it is not wholly OOD?

Why is some of the color coding and bolding not available for some cells in Tab. 1, 2? If it is to indicate that the cell’s result is conforming to the hypothesis, these details should be available in the Table’s caption

It is unclear where the empirical evidence for this is? Tab. 3 only shows the results for negative samples. If it is “richer” than positive samples we need to compare and contrast
> Negative samples exhibit a richer variety of reasoning patterns, whereas positive data tend to follow more consistent trajectories.

This can not be the right interpretation. If you Maximum likelihood estimate (MLE) on any given data, the train loss will fall.
> The negative curve remains monotonic and closely tracks the positive curve, indicating that negatives provide meaningful learning signals rather than noise

Why are sec. 5’s experimental results missing the Qwen2.5-32B and Llama3.1-8B present in previous section’s experiments?

There should be a limitation section, even if it is in the appendix due to space constraints.

**Questions:**

In what scenario is incorrect samples readily available other than distillation?
> us to inquire whether an SFT approach can not only improve data efficiency through utilizing all available samples but also benefit from this expanded exploration space to enhance generalization.

Since you claim that it explores more at inference time, it will be more convincing if you could try to do some simple test-time scaling like majority voting with N samples and see if your method helps.

Can the authors explain how
>• We provide the first systematic study demonstrating that negative reasoning samples constitute valuable supervision

Is true when you have an extended related work section on “Learning from Negative Data “

Can the authors explain in what real-world scenario we would distill Qwen3-8B → Qwen-2.5 14B, 32B ?

Are there any cost overheads for GLOW over the naive approach?

Has this work been compared with "Large Reasoning Models Learn Better Alignment
from Flawed Thinking" ?

---

> ### Author Response · Authors · 2025-11-21
> **Official Comment by Authors**
>
> We greatly appreciate your insightful feedback on our work! We have carefully considered all comments and questions, and have revised the paper accordingly to clarify our problem setting, better position our contributions relative to prior work, and strengthen both the empirical and theoretical support for our claims. Below, we respond to each point in detail and indicate the corresponding changes in the revised version.
>
> # Weakness 1
> > As the paper mentions, the pre-training + post-training (SFT, RL) is the common paradigm. There are clear evidence that shows that RL generalizes and SFT memorizes,e.g. “SFT Memorizes, RL Generalizes: A Comparative Study of Foundation Model Post-training” Modern post-training of reasoning models never stop at SFT, but always include the RL stage as it dramatically improves performance. Given that it is possible that RL improves generalization, for this work to be practically impactful it would need to show that the generalization gains at SFT lead to improved generalization or in-distribution performance post-RL.
>
> We appreciate the reviewer’s question regarding the practical relevance of improving SFT-stage generalization. Our motivation is not to suggest that SFT alone yields uniform gains across all tasks, but rather that the SFT stage remains a critical and highly effective component of modern LLM training. Empirically, SFT has consistently demonstrated strong sample efficiency, stable optimization behavior, and substantial improvements even without reinforcement learning. For example, [1] trains a reasoning agent using only SFT, without any RL.
>
> In practice, the effectiveness of downstream RL is shaped by several interacting factors: data quality, exploration dynamics, preference signal coverage, and, importantly, the model’s inherent reasoning capability before RL begins. A stronger SFT model, equipped with broader cross-domain generalization, typically explores more productively, discovers better candidate solutions, and avoids early collapse into narrow or brittle strategies. Improving the SFT foundation therefore enhances the “starting point” from which RL operates, even though quantifying this transfer precisely is outside the scope of the present study.
>
> Our contribution should therefore be understood as strengthening this foundation: we show that negative-sample supervision and our gain-based reweighting improve the generalization behavior of SFT itself, which in turn provides a more capable substrate for any subsequent preference optimization or RL-stage refinement. Extending this line of investigation to measure downstream effects on RLHF performance is a natural and important next step, and we view it as valuable future work beyond the boundaries of this paper.
>
> [1] https://x.com/Skywork_ai/status/1990816721251536918
>
> # Weakness 2
> >First line of the abstract is confusing. The phrasing makes it sound like SFT always uses CoT, which is incorrect. SFT with CoT is only for reasoning post-training.
>
> We thank the reviewer for pointing out this ambiguity in the abstract. To address this, we have revised the first sentence of the abstract as follows:
> > Supervised fine-tuning (SFT) on chain-of-thought (CoT) trajectories is a standard component of reasoning-oriented post-training for large language models. In current practice, such CoT-based SFT typically retains only trajectories whose final answers match the ground truth, which can lead to poor generalization due to overfitting and wasted data from discarding incorrect samples.
>
> This revision makes clear that:
> 1. We are discussing CoT-based reasoning SFT, not all SFT pipelines.
> 2. The “typically uses only correct CoT data” statement refers specifically to this reasoning post-training recipe, where rejection sampling on correctness is widely used.

---

> ### Author Response · Authors · 2025-11-21
> **Official Comment by Authors  (2)**
>
> # Weakness 3
> >Why is there + in the data annotation in Fig. 1? This seems unnecessary as it is not increasing; it is just the performance annotation.
>
> Thank you for raising this point. The “+X” annotations in Fig. 1 are not intended as redundant positive signs on raw performance values. Rather, they are used to explicitly mark the absolute improvement (in percentage points) achieved after SFT, so that the performance gains of positive-only and negative-only SFT can be visually distinguished more clearly. The aim is to highlight the magnitude of the improvement relative to the baseline model before SFT, not merely to restate the final accuracy.
> We agree that this intention was not clearly stated in the original caption, which could make the “+” appear unnecessary. To resolve this, we have updated the caption to state that:
> > Qwen2.5-14B trained with positive samples shows limited transfer beyond math, whereas models trained on negative samples generalize more broadly across reasoning tasks. \revadd{Bars report the final accuracy on each benchmark, and the “+” annotations above them denote the absolute accuracy improvement over the model before SFT.
>
> Revised caption provided in the updated PDF; see Fig. 1 in the revised submission.
>
> # Weakness 4
> > Typo issues
>
> We thank the reviewer for noticing these typos! We have corrected these in our revision.
>
> # Weakness 5
> > The communication is confusing. What does dominant SFT mean? Why is the idea of distillation suddenly introduced here?
> > (2) Data Inefficiency: Dominant SFT relies on distilled reasoning paths and uses rejection sampling to select those leading to correct answers and formats
>
> Thank you for pointing out this confusion.
> By “dominant SFT” we did not intend to introduce a new technical term, but to refer to the standard SFT-based reasoning post-training recipe that is widely used in current practice: starting from a pretrained LLM, one collects chain-of-thought reasoning traces from a stronger teacher model (e.g., Qwen or DeepseekV3.2) and then performs SFT on this distilled reasoning data, typically within a relatively specific reasoning domain (such as math or logic benchmarks) rather than broad multi-domain web corpora. We agree that our phrasing suggests that “Dominant SFT” is a well-defined concept, which makes the sentence harder to understand.
> Similarly, the mention of “distilled reasoning paths” in point (2) is not meant to suddenly introduce a new idea at that point, but simply to describe the data source underlying most recent reasoning SFT setups, where training trajectories are distilled from a stronger thinking model and then filtered via rejection sampling to keep only those with correct final answers and formats.
> To make this clearer, we will revise the corresponding sentence as follows:
> > (2) Data Inefficiency: Current reasoning post-training pipelines predominantly perform SFT on chain-of-thought trajectories distilled from a stronger teacher model, and use rejection sampling to retain only trajectories whose final answers and formats match the ground truth...
>
> # Weakness 6
> >This phrase is confusing. The communication needs revision.
> While relying solely on negative samples is a possible strategy, it remains a form of rejection sampling with low data efficiency
>
> Thank you for pointing this out. The sentence in line 89 was not intended to suggest that “negative-only SFT” is a realistic or recommended training setup. It was meant to provide a conceptual contrast that helps motivate our method. Our analysis shows that negative trajectories can contribute to generalization, but using only negatives would still amount to a rejection-sampling process, simply discarding positives instead of negatives. In that sense, both positive-only and negative-only SFT suffer from the same limitation because they filter out useful data. The core message we intended to convey is that achieving true data efficiency requires moving beyond rejection-based strategies and developing a way to make use of both correct and incorrect trajectories without discarding training signal. We have revised the text to make this intent clearer in the updated submission.
>
> # Weakness 7
> >It is not correct to call MMLU and MMLU-pro a common sense benchmark
>
> The original MMLU paper defines them as massive multi-task evaluations of knowledge and problem-solving across 57 academic and professional subjects, and follow-up work similarly treats MMLU as a broad knowledge and reasoning benchmark rather than a dedicated commonsense dataset. In the revised version, we have replaced “common sense benchmark” with “General Reasoning benchmarks”. This change does not affect any of our experiments or conclusions.

---

> ### Author Response · Authors · 2025-11-21
> **Official Comment by Authors (3)**
>
> # Weakness 8
> > MMLU includes mathematics so it is not wholly OOD?
>
> The point we intended to make is that the math data used for post-training and the math portions of MMLU come from substantially **different distributions**, not that there is no overlap in the broad subject category of “math.”
> Concretely, the OpenMathReasoning dataset is built from AoPS-style competition and olympiad problems with long, multi-step chain-of-thought solutions and tool-integrated reasoning traces, designed explicitly to train strong mathematical reasoning models. In contrast, MMLU math questions are short, four-choice multiple-choice items sourced from textbooks and standardized college-level exams, such as “In the complex plane, the set of points satisfying $z^2 = |z|^2$ is a (A)…(D)…”, which primarily test concise conceptual knowledge and exam-style problem solving rather than long-form solution construction.
> This mismatch in problem source (AoPS forums vs. textbooks/exams), format (open-ended CoT with long solutions vs. multiple choice), and difficulty profile (olympiad/contest vs. standard curriculum) makes the MMLU math subsets distributionally quite different from OpenMathReasoning, even though both are labeled as “math”.
>
> # Weakness 9
> > Why is some of the color coding and bolding not available for some cells in Tab. 1, 2? If it is to indicate that the cell’s result is conforming to the hypothesis, these details should be available in the Table’s caption
>
> Thank you for the comment. We have updated the captions to explicitly describe the meaning of both formats, ensuring that the tables are now self-contained. The revised version includes the corrected captions.
>
> # Weakness 10
> > It is unclear where the empirical evidence for this is? Tab. 3 only shows the results for negative samples. If it is “richer” than positive samples we need to compare and contrast
>
> We are delighted to further discuss this point. By “negative samples exhibit a richer variety of reasoning patterns”, we mean this in two concrete senses supported by our analysis.
> First, Table 3 quantifies error-type diversity on the negative side: following [1], we categorize each negative trajectory into 9 major error types and 22 sub-types using Gemini-2.5-Pro, and find that negative CoTs in OpenMathReasoning and MMLU are spread across a wide range of categories (calculation, completeness, logical, understanding, special cases, etc.). Positive trajectories, by construction, contain no explicit error segments to classify in the same way and are further constrained by the requirement to hit the exact ground-truth answer and a strict formatting template.
> Second, we complement this taxonomy view with a policy-entropy comparison between models trained on positives and negatives in Table 5. There, we analyze the entropy of the reasoning policy in the thinking span and answer span for $M_{pos}$ (positive-only SFT) and $M_{neg}$ (negative-only SFT). Across both math and “other” benchmarks, and under both off-policy and on-policy evaluation, $M_{neg}$ consistently shows higher entropy in the thinking span and a larger gap $\Delta H$ between thinking and answer entropy than $M_{pos}$. This indicates that negative-trained models explore a broader set of intermediate reasoning states before committing to an answer, which is consistent with their stronger cross-domain generalization. Intuitively, **a model can learn to explore a wide range of reasoning modes if such diversity is present in its training data, so the richer spectrum of negative trajectories provides a natural source for the more exploratory reasoning behaviour we observe.**
>
> [1] https://arxiv.org/abs/2502.19361

---

> ### Author Response · Authors · 2025-11-21
> **Official Comment by Authors (4)**
>
> # Weakness 11
> > This can not be the right interpretation. If you MLE on any given data, the train loss will fall.
> > The negative curve remains monotonic and closely tracks the positive curve, indicating that negatives provide meaningful learning signals rather than noise
>
> We thank the reviewer for this comment.
> Our intention in the original sentence was not to claim that the shape of the loss curve alone proves the usefulness of negative trajectories. **Rather, the observation that the negative-only SFT curve remains stable and closely tracks the positive-only curve was meant as a sanity check on optimization dynamics: training on negatives does not diverge or collapse, and be haves similarly to training on positives.**
> Beyond this optimization view, the evidence that negatives are genuinely useful comes from their generalization behaviour in Tables 1 and 2. In Table 1, where we train on the math reasoning dataset, negative-only SFT consistently maintains in-domain math performance at a level comparable to positive-only SFT across all backbones (the average accuracy on Math500 / Minerva / Olympia / AMC differs only marginally between the two). At the same time, on out-of-domain benchmarks the negative-only models show clear gains: on the common-knowledge/general-reasoning benchmarks (MMLU, MMLU-Pro, BBH) the improvements are on the order of 10 percentage points on average, and on other reasoning benchmarks (ACPBench, HeadQA) the gains are around 5 points. An analogous pattern appears in Table 2 when training on the MMLU-based general reason data: negative-only SFT preserves in-domain performance on general reasoning while still improving math and other reasoning benchmarks, again indicating that negatives help generalization rather than merely being fit on the training set.
>
> # Weakness 12
> > Why are sec. 5’s experimental results missing the Qwen2.5-32B and Llama3.1-8B present in previous section’s experiments?
>
> Thank you for raising this point. In the original paper, Section 5 focuses on more detailed ablations and analysis, and for computational reasons we restricted these experiments to the Qwen2.5 medium-scale backbones. Training the full set of Section 5 variants on Qwen2.5-32B is significantly more resource intensive, and the Llama3.1-8B backbone already underperforms Qwen2.5 at the base level in our setting, so we initially chose Qwen as the main model for validating our method and analyzing the phenomena.
> To address your concern about consistency across model families, we have **added experiments for both Qwen2.5-32B and Llama3.1-8B in the revised version** (see Table 6 and 7) and blow:
>
> |             | Training Data | Method |  math500  |  minerva  |  olympia  |    amc    |  Avg_math |    MMLU   |  MMLU-Pro |    BBH    | Avg_general |  ACPBench |   HeadQA  | Avg_other |
> |:-----------:|:-------------:|:------:|:---------:|:---------:|:---------:|:---------:|:---------:|:---------:|:---------:|:---------:|:-----------:|:---------:|:---------:|:---------:|
> | llama3.1-8B |      MATH     |   SFT  |   41.20   |   18.01   |   14.67   |   15.00   |   22.22   |   62.48   |   36.88   |   55.12   |    51.49    |   32.96   |   65.90   |   49.43   |
> |             |               |  GLOW  | **44.60** | **20.59** | **15.11** | **17.50** | **24.45** |  **63.80** | **38.34** | **58.17** |  **53.44**  | **35.04** | **66.70** | **50.87** |
> |             |      MMLU     |   SFT  |   20.00   |   15.81   |    6.52   |    2.50   |   11.21   |   66.49   |   40.56   |   53.73   |    53.59    |   36.06   |   69.55   |   52.81   |
> |             |               |  GLOW  | **24.80** | **20.59** |  **6.96** | **12.50** | **16.21** | **68.52** | **42.95** | **57.53** |  **56.33**  | **39.72** | **72.57** | **56.15** |
> | Qwen2.5-32B |      MATH     |   SFT  |   92.20   |   52.57   |   57.19   |   85.00   |   71.74   |   85.22   |   73.10   |   83.53   |    80.62    |   50.67   |   84.90   |   67.79   |
> |             |               |  GLOW  | **93.40** | **54.41** | **59.11** | **92.50** | **74.86** | **85.51** | **74.14** | **83.98** |  **81.21**  | **51.97** | **85.19** | **68.58** |
> |             |      MMLU     |   SFT  |   86.60   |   46.69   |   55.70   |   80.00   |   67.25   |   79.06   |   61.15   |   79.94   |    73.38    |   49.89   |   83.01   |   66.45   |
> |             |               |  GLOW  | **89.00** | **47.06** | **58.67** | **82.50** | **69.31** | **80.81** | **64.72** | **81.98** |  **75.84**  | **52.08** | **83.73** | **67.91** |
>
> These additional results confirm that the phenomena analyzed in Section 5 are not specific to a single backbone size or family, but also hold for the larger Qwen2.5-32B and the Llama3.1-8B models.

---

> ### Author Response · Authors · 2025-11-21
> **Official Comment by Authors (5)**
>
> # Weakness 13
> > There should be a limitation section, even if it is in the appendix due to space constraints.
>
> We add limitation in the revised version.
>
> # Q1
> > In what scenario is incorrect samples readily available other than distillation?
>
> We appreciate the reviewer’s question. In our experiments, incorrect trajectories happen to come from a distillation pipeline, but we do not view distillation as the only or even the primary way such negatives arise in practice. Whenever a reasoning model is used in a ReAct[1] loop, incorrect samples are naturally and abundantly produced.
> Concretely, there are at least three common scenarios.
>
> First, RLHF and preference-learning pipelines routinely sample multiple candidate responses per prompt and ask humans to rank them; all lower-ranked or rejected candidates are effectively “incorrect” trajectories that are currently discarded or only used implicitly through pairwise preferences.
>
> Second, self-correction and multi-turn RL methods generate entire trajectories of failed attempts and subsequent revisions using only the model itself; these unsuccessful attempts again constitute exactly the kind of negative chain-of-thought data we study.
>
> Third, deployed chatbots and tutoring systems continuously log responses that users mark as wrong or revise in follow-up turns, providing large volumes of noisy but readily available incorrect samples, especially in domains with automatic checkers such as math or code.
>
>
> [1] https://arxiv.org/abs/2210.03629
>
> # Q2
> > Since you claim that it explores more at inference time, it will be more convincing if you could try to do some simple test-time scaling like majority voting with N samples and see if your method helps.
>
> We appreciate the reviewer’s suggestion. We view test-time scaling with multiple samples as **orthogonal** to our main contribution, and claim that the model “explores more at inference time” is grounded in the analysis we already report under a fixed single-sample decoding protocol: the reweighted models exhibit higher entropy on the reasoning head, more diverse chain-of-thought trajectories, and, crucially, improved performance on both in-domain and out-of-domain benchmarks compared to standard SFT, even when using the same decoding strategy as prior work. In other words, our primary goal in this paper is to show that gain-based reweighting leads to a better base policy under standard inference settings, without relying on additional test-time tricks. To keep the experimental design focused and comparable to existing SFT-based reasoning work, we therefore restrict our reported results to the single-sample setting and leave a systematic study of self-consistency and other test-time scaling strategies on top of our reweighted policy as future work.
>
> # Q3
> > Can the authors explain how
> > • We provide the first systematic study demonstrating that negative reasoning samples constitute valuable supervision
>
> Our claim is not that we are the first to use negative data in general, but that we are the first to systematically treat negative chain-of-thought trajectories as an independent **useful and informative** source of  supervision for reasoning SFT. Ref to line 150-152 in the revised paper.

---

> ### Author Response · Authors · 2025-11-21
> **Official Comment by Authors (6)**
>
> # Q4
> > What about comparison or discussion with "The Surprising Effectiveness of Negative Reinforcement in LLM Reasoning"?
>
> We thank the reviewer for pointing out this work. At a high level, both [1] and our paper highlight that “negative” reasoning samples carry important learning signal, but they do so in different parts of the post-training pipeline and under different training regimes: [1] studies negative reinforcement in an RL with verifiable rewards (RLVR) setting, whereas we study negative chain-of-thought supervision in supervised fine-tuning (SFT) and how to reweight it for better generalization.
>
> **First, our work operates at the SFT stage, while [1] operates at the RL stage.** We focus on SFT on distilled chain-of-thought trajectories and analyze how positive-only, negative-only, and mixed SFT affect in-domain and out-of-domain generalization across math and multi-task knowledge benchmarks. In contrast, [1] decomposes RLVR updates into positive-sample reinforcement (PSR) and negative-sample reinforcement (NSR) and empirically compares these RL variants on math benchmarks.
>
> **Second, we treat incorrect CoT trajectories as direct supervision, whereas [1] primarily uses negative samples as penalties.** Our method learns from negative trajectories themselves by feeding them into an SFT loss with gain-based reweighting, letting training decide when negatives should be upweighted or downweighted. By comparison, [1] shows that NSR-only training, which penalizes incorrect responses without explicitly reinforcing correct ones, can significantly improve Pass@k and sometimes match or surpass PPO/GRPO, but negatives there enter as a reward shaping signal rather than as supervised CoT targets.
>
> Conceptually, the two results are complementary. [1] shows that penalizing incorrect trajectories in RL can sharpen a reasoning policy without sacrificing Pass@k diversity, while we show that training directly on negative CoT trajectories in SFT can preserve in-domain accuracy and improve out-of-domain generalization.
>
> [1] https://arxiv.org/pdf/2506.01347
>
> # Q5
> > Can the authors explain in what real-world scenario we would distill Qwen3-8B → Qwen-2.5 14B, 32B ?
>
> Thank you for the question. Concretely, we treat Qwen3-8B as a compact reasoning specialist that already has strong chain-of-thought capability (e.g., after expensive CoT distillation or RL on math/logic), and use it as a teacher to transfer this reasoning ability into larger, more general-purpose backbones like Qwen2.5-14B/32B that have not been explicitly trained for reasoning. In many practical systems it is prohibitively costly to re-run full reasoning-oriented training on every large backbone, so distilling the small specialist’s trajectories into larger students is a natural way to endow them with strong reasoning skills at much lower cost. This “strong reasoner → larger non-reasoner” distillation setup precisely matches the weak-to-strong reasoning (W2SR) scenario (e.g., [1]): a relatively weaker/smaller teacher is used to unlock and amplify latent reasoning ability in a stronger student, without having to directly supervise the large model with equally strong labels.
>
> [1] https://arxiv.org/pdf/2505.20072
>
> # Q6
> > Are there any cost overheads for GLOW over the naive approach?
>
> Yes, GLOW does introduce a small overhead compared to the naïve SFT baseline, but it is limited to lightweight bookkeeping during training. In our implementation, GLOW keeps a dictionary that stores a few scalar statistics per training example (e.g., the current gain estimate keyed by sample ID). At each step we only perform a hash lookup and an in-place update for the samples in the current mini-batch, and then compute a scalar weight that multiplies the standard SFT loss. There are no additional forward or backward passes and no changes to the model architecture.

---

> ### Author Response · Authors · 2025-11-21
> **Official Comment by Authors (7)**
>
> # Q7
> > Has this work been compared with "Large Reasoning Models Learn Better Alignment from Flawed Thinking" ?
>
> The first arXiv version of “Large Reasoning Models Learn Better Alignment from Flawed Thinking” was posted on October 1, 2025, which is **after the ICLR 2026 submission deadline** on September 24, 2025, so we could not compare against or cite it in the original submission. We will add a discussion and citation for this work in the revised version.
>
> Conceptually, this work studies flawed chain-of-thought at the RL stage for safety alignment: RECAP is an RLHF-style method that uses counter-aligned CoT prefills and a verifiable safety reward to teach models to override unsafe reasoning and improve jailbreak robustness. In contrast, our work operates purely at the SFT stage for general reasoning, treating incorrect CoT trajectories as supervision for math and multi-task benchmarks, and proposing a gain-based reweighting scheme that improves out-of-domain generalization under standard single-sample decoding.

---

> ### Author Response · Authors · 2025-11-25
> **Request for Feedback Before Discussion Period Ends**
>
> Dear Reviewer SmfD:
>
> As the author-reviewer discussion period approaches its end, we kindly request your feedback on our rebuttal. Your insights on whether we have effectively addressed your concerns would be greatly appreciated.
>
> We are truly grateful for the time and expertise you have dedicated to reviewing our work. Your thoughtful comments and suggestions have significantly contributed to improving the quality of our research.
>
> Thank you once again for your valuable feedback. We look forward to your guidance and hope to address any remaining concerns before the discussion period closes.
>
> Yours Sincerely,
>
> All Authors

---

> ### Comment · Reviewer_SmfD · 2025-11-26
>
> Many thanks to the authors that put effort towards the rebuttal phase.
>
> After a look at the revised version, the communication and presentation still remains quite challenging to follow.s
>
> It has been shown in this paper "Quagmires in SFT-RL Post-Training: When High
> SFT Scores Mislead and What to Use Instead" that standard performance metric is poorly correlated with post-RLVR performance. Instead using large pass@k for your evals will be a very simple and quick test you can do for this weakness.
>
> > A stronger SFT model, equipped with broader cross-domain generalization, typically explores more productively, discovers better candidate solutions, and avoids early collapse into narrow or brittle strategies. Improving the SFT foundation therefore enhances the “starting point” from which RL operates, even though quantifying this transfer precisely is outside the scope of the present study.
>
> It is unclear why this simple test time evaluation cannot be made. For similar reasons stated above. This is quite important for post-RLVR stage. For instance your experiments and your response suggests that this is reasoning distillation from qwen 3 --> qwen 2. if this is the case, in a standard industry setting we would do RLVR after this SFT/distillation stage.
>
> > Since you claim that it explores more at inference time, it will be more convincing if you could try to do some simple test-time scaling like majority voting with N samples and see if your method helps.
>
> If this method is distillation-specific it should be framed this way. The title/writing should also follow this to make the scope of this paper clear.
>
> This explanation is not convincing. For ReAct its self-made incorrect samples, which was never studied in this paper. this paper's evidance is strictly for a better (reasoning) model --> worse model, as the authors state; not for self-distillation as is the case would be if this was implemented for ReAct. Same goes for the second and third point where the setting is different from the experimental setting. For instance, RLHF style positive and negative labels are highly different from the CoT-style reasoning, and SFT experiment that you showed.
>
> > We appreciate the reviewer’s question. In our experiments, incorrect trajectories happen to come from a distillation pipeline, but we do not view distillation as the only or even the primary way such negatives arise in practice. Whenever a reasoning model is used in a ReAct[1] loop, incorrect samples are naturally and abundantly produced.
> Concretely, there are at least three common scenarios.
>
> > First, RLHF and preference-learning pipelines routinely sample multiple candidate responses per prompt and ask humans to rank them; all lower-ranked or rejected candidates are effectively “incorrect” trajectories that are currently discarded or only used implicitly through pairwise preferences.
>
> > Second, self-correction and multi-turn RL methods generate entire trajectories of failed attempts and subsequent revisions using only the model itself; these unsuccessful attempts again constitute exactly the kind of negative chain-of-thought data we study.
>
> > Third, deployed chatbots and tutoring systems continuously log responses that users mark as wrong or revise in follow-up turns, providing large volumes of noisy but readily available incorrect samples, especially in domains with automatic checkers such as math or code.
>
> The revised ver.'s communication needs to make sure this is fixed.
>
> >This can not be the right interpretation. If you MLE on any given data, the train loss will fall. The negative curve remains monotonic and closely tracks the positive curve, indicating that negatives provide meaningful learning signals rather than noise
>
> Some of the weaknesses that were raised have been resolved, and in response I am raising my score 2 --> 4 and confidence to 4. Nevertheless, this work would benefit from a **major thorough revision** on weaknesses that myself and other reviewers raised.

---

> > ### Author Response · Authors · 2025-11-27
> > **Official Comment by Authors (8)**
> >
> > # Q1-1
> > We respectfully emphasize that our paper is exclusively about the SFT stage of post-training, focusing on how learning from negative reasoning trajectories reshapes the optimization landscape, improves conditioning, and **enhances cross-domain generalization** before any RL or RLVR stage is applied.
> > Our work aims to answer a specific and self-contained question: *Can incorrect reasoning trajectories be leveraged as useful supervision signals during SFT, even before any RL or preference optimization is introduced?*
> > This is fundamentally an SFT-level scientific question, and all our theoretical analysis, empirical setups, and ablations are designed for this setting.
> >
> > Nonetheless, during the rebuttal period we also ran small RL experiments with verl to directly verify that our approach is compatible with and beneficial to RL-style post-training . We fine-tuned Qwen2.5-7B-base on GSM8K under four settings (SFT, SFT+RL, GLOW, GLOW+RL), keeping all RL-stage hyperparameters, data, and optimization details strictly identical and only varying whether the preceding SFT stage used our GLOW reweighting. Compared to the standard SFT baseline, our method yields consistent gains both before and after RL. At the SFT stage, GLOW improves MMLU from 72.24 to 73.99 (+1.75 points) and MMLU-Pro from 53.71 to 55.77 (+2.06 points), for an average gain of +1.91 points across these two OOD benchmarks. After applying the same RL procedure, GLOW+RL further improves MMLU from 72.82 to 76.47 (+3.65 points) and MMLU-Pro from 53.95 to 57.37 (+3.42 points), averaging +3.54 points. This shows that our advantage still persists after RL phase, indicating improved initialization that remains beneficial under RL-style post-training.
> >
> > |            |         | GSM8K(test) | Math500  | Minerva  | Olympia  | AMC  |  MMLU | MMLU-Pro |
> > |------------|---------|-------------|:-------:|:--------:|:--------:|:--------:|:-----:|:--------:|
> > | Qwen2.5-7B | SFT     | 89.20       | 76.60   | 40.07    | 38.96    | 55.00    | 72.24 | 53.71    |
> > |            | SFT+RL  | 92.58       | 78.20   | 40.24    | 39.67    | 52.50    | 72.82 | 53.95    |
> > |            | GLOW    | 89.80       | 79.60   | 40.07    | 41.04    | **60.00**    | 73.99 | 55.77    |
> > |            | GLOW+RL | **93.06**       |**80.20**   | **42.28**    | **41.78**    | 57.50    | **76.47** | **57.37**    |

---

> > ### Author Response · Authors · 2025-11-27
> > **Official Comment by Authors (9)**
> >
> > # Q1-2
> > We agree with this perspective: single-sample accuracy is an imperfect proxy for post-training behavior, and test-time scaling measured via pass@k provides a more informative signal. To address your concern, we are currently running pass@4/8/16/32 evaluations and have allocated all available GPUs to generate sufficient samples. Due to limited time and computational resources, we report results on Math500 and Minerva. We evaluate Qwen2.5-7B models fine-tuned for general reasoning, respectively, using either positive or negative samples.
> >
> > ### Math500 (Pass@k)
> > | Model        | Pass@4 | Pass@8 | Pass@16 | Pass@32 |
> > |--------------|:------:|:------:|:-------:|:-------:|
> > | positive  | 85.38  | 88.41  | 90.56   | 92.00   |
> > | negative  | 87.60  | 89.61  | 91.65   | 93.20   |
> >
> > ### Minerva (Pass@k)
> > | Model        | Pass@4 | Pass@8 | Pass@16 | Pass@32 |
> > |--------------|:------:|:------:|:-------:|:-------:|
> > | positive  | 47.78  | 52.35  | 56.44   | 60.29   |
> > | negative  | 49.40  | 54.53  | 59.16   | 62.87   |
> >
> >
> > Overall, **models trained on negative samples consistently outperform their positive-sample counterparts across all values of k in OOD benchmarks**. Together with Table 5 in the paper, these results indicate that negative samples provide a meaningful training signal, encourage broader exploration during learning, and **constitute a strong starting point for subsequent RL post-training**.

---

> > ### Author Response · Authors · 2025-11-27
> > **Official Comment by Authors (10)**
> >
> > # Q2
> > We thank the reviewer for this helpful feedback. We will explicitly note that **distillation is not a limitation in practice: it is a simple, scalable, and effective way** to obtain both positive and negative reasoning trajectories with consistent formatting and controllable quality, making it a practical choice for studying negative supervision in standard SFT pipelines. **We validate the effectiveness of SFT on distilled trajectories**. Moreover, as shown by the additional results in Q1, using **our method as an initialization for RL post-training yields better performance**, further supporting the practicality of this simple pipeline.
> >
> > Importantly, the empirical advantage of negative samples serves primarily as motivation and as a basis for analysis. **Our method does not depend on the sample type** (positive vs. negative); it depends on the $\Delta loss$ of each sample across epochs to perform dynamic reweighting. This continuously redirects training toward regions the model has not yet adequately explored, instead of merely improving performance on negative samples. Accordingly, it outperforms training with only positive samples, only negative samples, or full mixture.
> >
> > Besides, we revise the paper to more clearly emphasize that our setting is specifically about using distilled reasoning trajectories in the SFT stage (see line 43–45), and to avoid any possible overinterpretation beyond this scenario.

---

> > ### Author Response · Authors · 2025-11-27
> > **Official Comment by Authors (12)**
> >
> > # Q3-2
> > ### Qwen2.5-32B — **Math reasoning** (Positive vs. Negative distilled trajectories)
> > | Epoch  | Trajectory | Math500 | Minerva | Olympia |  AMC  |  MMLU | MMLU-Pro |  BBH  |
> > |-------:|-----------:|:-------:|:-------:|:-------:|:-----:|:-----:|:--------:|:-----:|
> > | Base   | —          | 63.20   | 34.19   | 26.52   | 35.00 | 68.34 | 39.80    | 58.65 |
> > | 5epoch | Positive   | 90.20   | 49.63   | 59.11   | 85.00 | 76.53 | 46.77    | 78.04 |
> > |        | Negative   | 88.40   | 45.22   | 52.30   | 85.00 | **83.07** | **68.23**    | **83.55** |
> > | 10epoch| Positive   | 92.60   | 50.00   | 60.44   | 85.00 | 78.63 | 51.67    | 79.01 |
> > |        | Negative   | 92.20   | 51.10   | 57.93   | 85.00 | **85.14** | **73.75**    | **84.22** |
> > | 15epoch| Positive   | 93.00   | 48.53   | 62.07   | 90.00 | 78.72 | 51.99    | 80.57 |
> > |        | Negative   | 91.20   | 50.74   | 57.33   | 90.00 | **85.02** | **73.48**    | **84.62** |
> > | 20epoch| Positive   | 91.40   | 50.74   | 60.89   | 85.00 | 79.01 | 54.31    | 80.61 |
> > |        | Negative   | 92.20   | 50.74   | 58.37   | 95.00 | **85.47** | **73.53**    | **84.51** |
> >
> > ### Qwen2.5-32B — **General reasoning** (Positive vs. Negative distilled trajectories)
> > | Epoch  | Trajectory | Math500 | Minerva | Olympia |  AMC  |  MMLU | MMLU-Pro |  BBH  |
> > |-------:|-----------:|:-------:|:-------:|:-------:|:-----:|:-----:|:--------:|:-----:|
> > | Base   | —          | 63.20   | 34.19   | 26.52   | 35.00 | 68.34 | 39.80    | 58.65 |
> > | 5epoch | Positive   | 84.60   | 44.85   | 52.00   | 62.50 | 82.10 | 66.54    | 80.03 |
> > |        | Negative   | **85.00**   | 44.49   | 51.26   | **77.50** | 78.74 | 57.48    | 79.09 |
> > | 10epoch| Positive   | 86.60   | 46.69   | 55.70   | 75.00 | 81.14 | 67.01    | 80.69 |
> > |        | Negative   | **87.20**   | 46.30   | 54.52   | **75.00** | 79.01 | 60.43    | 80.88 |
> > | 15epoch| Positive   | 85.00   | 47.06   | 56.59   | 75.00 | 81.73 | 68.33    | 81.73 |
> > |        | Negative   | **86.40**   | **47.79**   | 55.70   | 65.00 | 77.77 | 57.14    | 79.97 |
> > | 20epoch| Positive   | 85.20   | 46.69   | 56.15   | 75.00 | 81.97 | 68.54    | 81.60 |
> > |        | Negative   | **86.40**   | **47.06**   | **56.89**   | 72.50 | 77.99 | 58.34    | 80.71 |
> >
> > From the checkpoint evaluations during training, we observe that learning from negative samples consistently outperforms learning from positive samples on out-of-domain benchmarks. This suggests that negative trajectories provide a more transferable training signal, leading to better generalization.

---

> > ### Author Response · Authors · 2025-11-27
> > **Official Comment by Authors (13)**
> >
> > We would like to restate our key contributions clearly.
> > 1. First, to the best of our knowledge, we are the first to empirically demonstrate that optimizing SFT toward negative reasoning trajectories yields consistent performance gains.
> > 2. Second, motivated by this finding, we propose GLOW, a simple dynamic reweighting method that improves data utilization without any manual data filtering.
> >
> > Across our experiments, GLOW improves performance under SFT and also provides a stronger initialization for subsequent RL post-training, leading to better generalization. Following your suggestions, we have added additional experiments and analyses to strengthen the evidence, and we hope they address your concerns. If anything remains unclear, we would be happy to clarify and further improve the paper.

---

> ### Author Response · Authors · 2025-11-27
> **Official Comment by Authors (11)**
>
> # Q3-1
> Thank you for the positive response to our earlier clarification. In the revised manuscript, we have added a more explicit explanation of the loss-curve behavior under MLE training (see line 315-319). In addition, to more directly demonstrate that the model is indeed learning from the negative trajectories, **we have included new evaluations of Qwen2.5-7B and Qwen2.5-32B at 5-epoch, 10-epoch, and 15-epoch checkpoints to show there progress performance during training on positive and negative samples (as shown below)**. These results, together with the observation that the loss curves for negative and positive trajectories follow similar trends during training, provide complementary evidence that the negative samples are not noise. We have also included this in the revised manuscript (see Table 9).
>
>
> ### Qwen2.5-7B — **Math reasoning** (Positive vs. Negative distilled trajectories)
> | Epoch  | Trajectory | Math500 | Minerva | Olympia |  AMC  |  MMLU | MMLU-Pro |  BBH  |
> |-------:|-----------:|:-------:|:-------:|:-------:|:-----:|:-----:|:--------:|:-----:|
> | Base   | —          | 58.40   | 26.84   | 26.07   | 52.50 | 55.80 | 26.56    | 51.10 |
> | 5epoch | Positive   | 72.80   | 37.13   | 37.19   | 45.00 | 60.95 | 30.34    | 54.69 |
> |        | Negative   | 71.20   | 31.99   | 31.56   | 47.50 | **62.58** | **44.04**    | **56.28** |
> | 10epoch| Positive   | 75.80   | 38.24   | 40.59   | 65.00 | 64.06 | 32.50    | 61.62 |
> |        | Negative   | 77.20   | 34.93   | 39.26   | 50.00 | **71.39** | **52.14**    | **69.49** |
> | 15epoch| Positive   | 77.20   | 36.76   | 41.93   | 55.00 | 60.81 | 32.15    | 59.69 |
> |        | Negative   | 78.60   | 39.71   | 38.37   | 52.50 | **72.10** | **52.24**    | **71.09** |
> | 20epoch| Positive   | 78.00   | 36.76   | 41.78   | 57.50 | 61.03 | 32.70    | 60.58 |
> |        | Negative   | 77.60   | 40.44   | 38.37   | 57.50 | **73.11** | **53.74**    | **71.73** |
>
> ### Qwen2.5-7B — **General reasoning** (Positive vs. Negative distilled trajectories)
> | Epoch  | Trajectory | Math500 | Minerva | Olympia |  AMC  |  MMLU | MMLU-Pro |  BBH  |
> |-------:|-----------:|:-------:|:-------:|:-------:|:-----:|:-----:|:--------:|:-----:|
> | Base   | —          | 58.40   | 26.84   | 26.07   | 52.50 | 55.80 | 26.56    | 51.10 |
> | 5epoch | Positive   | 72.00   | 36.76   | 37.33   | 47.50 | 73.62 | 50.61    | 64.05 |
> |        | Negative   | **76.80**   | **36.76**   | **37.78**   | **47.50** | 71.09 | 43.99    | 66.00 |
> | 10epoch| Positive   | 74.60   | 37.50   | 41.48   | 55.00 | 73.79 | 53.32    | 69.73 |
> |        | Negative   | **76.80**   | **37.87**   | 40.30   | 52.50 | 71.43 | 45.87    | 68.84 |
> | 15epoch| Positive   | 72.00   | 37.50   | 39.26   | 50.00 | 74.11 | 53.91    | 68.34 |
> |        | Negative   | **76.80**   | **37.13**   | **41.48**   | **55.00** | 71.30 | 44.62    | 69.30 |
> | 20epoch| Positive   | 74.40   | 37.50   | 39.85   | 50.00 | 73.42 | 53.22    | 68.23 |
> |        | Negative   | **77.00**   | 37.13   | **42.07**   | **60.00** | 71.23 | 45.79    | 69.46 |

---

### Official Review · Reviewer_ZzKP · 2025-10-28

**Soundness:** 3
**Presentation:** 3
**Contribution:** 3
**Rating:** 6
**Confidence:** 3

**Summary:**

This paper investigates whether incorrect reasoning samples, usually discarded in supervised fine-tuning (SFT) of LLMs, can enhance generalization. The authors show that negative samples introduce diverse reasoning patterns, slow overfitting, and increase exploration during inference, leading to significant out-of-domain (OOD) gains. They further propose GLOW, which adaptively emphasizes underexplored samples, achieving consistent improvements across tasks.

**Strengths:**

(1) This paper presents a novel perspective by treating incorrect reasoning trajectories as useful supervision.

(2) The findings that incorrect reasoning samples are useful supervision are insightful, providing a potential direction for improving OOD generation in LLMs

(3) The writing is well-structured and easy to follow.

(4) Both empirical evidence and theory analysis are provided.

**Weaknesses:**

[1] The claim that this is “the first systematic study demonstrating that negative reasoning samples constitute valuable supervision” may be overstated, as similar claims have been made in prior work (e.g., [1]).

[2] The experimental evaluation is limited in scope, as most results are based on Qwen models. It remains unclear whether GLOW would provide consistent improvements across other architectures such as GPT-OSS-20B or LLaMA.


[1] Xinyu Zhu et al. The Surprising Effectiveness of Negative Reinforcement in LLM Reasoning. Nuerips 2025.

**Questions:**

(1)Could the authors clarify whether different types of reasoning errors contribute differently to supervision? In particular, how are supervision signals distributed among the various error categories, and do some error types provide more benefit than others?

(2) Can the authors comment on whether GLOW achieves consistent improvements on other architectures, such as GPT-OSS-20B?

---

> ### Author Response · Authors · 2025-11-21
> **Official Comment by Authors**
>
> # Weakness 1
> We thank the reviewer for pointing us to [1]. In short, [1] aims to suppress negative trajectories, whereas our method aims to learn from them.
> Difference from [1] in how negative samples are used. Our work and [1] both highlight the value of incorrect trajectories, but they use them for different objectives in different ways. [1] operates in an RL setting and uses negative samples as negative reinforcement signals that reduce probability mass in those directions. In contrast, our method stays in the SFT regime: negative trajectories are treated as additional supervision, because many of them contain partially correct or diverse reasoning patterns that help improve robustness and out-of-domain generalization. GLOW reweights such samples based on learning dynamics rather than outcome correctness, providing soft, structure-aware weighting rather than RL-style repulsion. Thus, both works exploit information in incorrect trajectories, but [1] uses them as repulsive signals under a reward-maximization objective, whereas we use them as diverse training signals under a generative modeling objective.
>
> [1] Xinyu Zhu et al. The Surprising Effectiveness of Negative Reinforcement in LLM Reasoning. Nuerips 2025.
>
> # Weakness 2 & Q2
> We appreciate the opportunity to clarify this point. GLOW is architecture-agnostic, and to demonstrate its applicability we conducted experiments on both Qwen and LLaMA, observing consistent trends across these backbones. We chose these models because the open-source community predominantly relies on LLaMA and Qwen, while GPT-OSS-20B was released relatively recently. Given our limited training resources, we were not able to include MoE architectures in this work, but we view extending GLOW to MoE models as an important direction for future work.
>
> # Q1
> This is an excellent question and one we have thought about carefully. In our view, the gains do not come from a specific type of data per se, but from the diversity of errors. Compared with fully correct samples, incorrect ones expose the model to a wider range of reasoning paths. This broadens the space of trajectories the model explores during thinking, which in turn leads to better performance. Thus, the key is not one particular category of mistakes, but having many different kinds of mistakes that enrich the training signal.

---

### Official Review · Reviewer_4qLV · 2025-10-31

**Soundness:** 2
**Presentation:** 2
**Contribution:** 2
**Rating:** 4
**Confidence:** 2

**Summary:**

The paper starts by observing and analyzing that negative reasoning traces are important. Then, based on these observations, the authors propose a GLOW mechanism that adaptively reweights the loss. Its efficacy is shown for various tasks.

**Strengths:**

- Well-written
- Interesting observations regarding negative reasoning traces, and a relatively simple proposed fix via loss-based reweighting
- Promising preliminary empirical results

**Weaknesses:**

- Theoretical discussions and statements in the main text and Appendix A.5 should be cleaned up:
   - In Eqn. (7), the index $t$ is missing rom LHS and $\ell_i^{(t)}$ should be just $\ell_i$? I guess that this is the training objective per epoch? So then $t$ is the epoch, not the individual (SGD) step?
   - What is $\delta$ in line 404?
   - Please utilize appropriate amsthm environments (lemma, theorem, etc). Also, theorem statements should not contain parts of the proof.
   - (A4) is not really an assumption.
   - What is the justification of the energy condition for Lemma 3?
   - In Lemma 3, why suddenly introduce a low-dimensional subspace?
   - As the main text promised rigorous statements, they should be included! For instance, Lemma 5 simply refers to Bousquet & Elisseeff (2002); Hardt et al. (2016), which makes the paper not so much self-contained..
   - The authors claim that the proposed reweighting scheme makes the loss more well-conditioned, per epoch. However, considering that the "true" objective is $\sum_i \ell_i$, the correct phrase is that the gradient direction is the gradient of a well-conditioned pseudo-loss per iteration. Of course, as the weights change, so does the minimum. Thus, as in [1, Theorem 2], a meaningful theoretical contribution may be to provide some convergence guarantee under this dynamic reweighting scheme.
- The proposed approach is, naturally, quite closely related to prior loss-based reweighting schemes ([2,3] and references therein). Yet, it is not clear what the advantages of the proposed reweighting are from this paper over the prior works and why this is particularly well-suited for LLM reasoning task beyond other ML tasks where reweighting has seen much success.


[1] https://openreview.net/forum?id=gU4ZgQNsOC

[2] https://arxiv.org/abs/2408.09849

[3] https://openreview.net/forum?id=GLUIuli3Sm

**Questions:**

1. I'm a bit confused on the training objective. So, per (ques, ans) pair, Qwen3-8B is used to sample a (reasoning, sampled-ans) conditioned on ques? Then the authors train on $\log \pi(reasoning, sampled-ans | ques)$, *regardless* of the correctness of the sampled-ans? Intuitively (e.g., as in [4]), at least I thought that this would bias the model to generate incorrect responses for the training set...? Here, $\pi$ is the model used for SFT, such as Qwen2.5 or Llama-3.1.

2. Recent work [4] considers the effect of negative reinforcement in LLM for RL (PPO and GRPO). Of course one clear difference is that they consider RL while this work considers SFT. Still,

3. I didn't quite understand why the diversity of incorrect reasonings is related/understood via IRM. What is the "environment" here? When doing next-token prediction, what is the shared representation $\Phi$ and predictor $w$ in our scenario? As the authors claim part of their contribution to be "deep analysis of how negatives improve generalization", I feel that this should be clarified and elaborated much more.

4. In Figure 1(b), is the plotted loss the same total training loss? What I mean is, is the blue and red curves both tracking $\sum_i \ell_i(w_t)$, where $w_t$ either tracks the AdamW of positive loss or negative loss?

5. Beyond SFT, would a similar approach work for self-improvement-style algorithms like STaR? According to this paper, am I correct in understanding that filtering is not necessary? This seems somewhat contradictory to many REINFORCE-style algorithms, where in verifiable reward setting (e.g., $0$ or $1$), the gradient is weighted by the indicator reward. I guess here, some discussions related to [4] would be appreciated as well.

6. Personally, I've never seen the stepwise loss decay as shown in all the plots with only cosine scheduling (as written in Appendix A.1). This seems like a stepwise schedule...? Is it usual?

7. (minor) Is it natural that the full fine-tuning is the only baseline?

8. Is it safe to claim that "training solely on negative samples outperform those trained on positive samples" based on the paper's experiments?

9. (Minor) Very recently, there have been some issues regarding floating-point operations (https://x.com/QPHutu/status/1984258808332550245). Can the authors comment on whether this issue(?) applies to their experiments?

[4] https://openreview.net/forum?id=ftVlLG9cks (arXiv ver1 came out on June, and so I don't think that it is unreasonable to bring this up)

---

> ### Author Response · Authors · 2025-11-21
> **Official Comment by Authors**
>
> Thank you for your positive assessment of our work and for your detailed comments. We greatly appreciate these high-quality suggestions and would be very glad to discuss them further.
> # Weakness 1
> **Discussion of Eq. (7) and the theoretical proof**
>
> - In Eq. (7),
>   $
>   L_{\text{GLOW}}^{(t)}(\theta)=\sum_{i=1}^N \omega_i^{(t)}\ell_i^{(t)},
>   $
>   where $t$ indexes the training epoch, $\ell_i^{(t)}$ is the loss of sample $i$ at epoch $t$, $\omega_i^{(t)}$ is the cross-epoch reweighting factor for that sample, and $L_{\text{GLOW}}^{(t)}(\theta)$ is the sum of reweighted losses over all samples in epoch $t$.
>
> - At line 404 there is a typo: the correct term is
>   $
>   \Delta w_i g_i g_i^\top,
>   $
>   not
>   $
>   \delta w_i g_i g_i^\top.
>   $
>   We apologize for the confusion. This has been corrected in the revised version.
>
> - **Why do we introduce the subspace $U$?**
>   We are specifically interested in improving curvature along directions with small eigenvalues. The subspace $U$ is the span of these ill-conditioned directions and is a standard tool in conditioning analysis, because it isolates the “worst” part of the spectrum on which we want to show improvement.
>
> - **Why do we introduce the energy condition?**
>   The reweighting rule naturally increases gradient energy along directions that are underrepresented, in particular those associated with small eigenvalues. The energy condition formalizes this effect and ensures that the additional mass introduced by reweighting indeed acts on the subspace $U$, so that curvature in these directions is provably enhanced.
>
> - **Comparison with [1]**
>   Compared to [1], our work differs substantially in the reweighting signal, problem setting, and theoretical goal. [1] proposes a loss-based reweighting scheme for large-scale pretraining: it assigns weights to training examples according to their instantaneous loss and analyzes, under convex and interpolation assumptions, how this improves convergence guarantees for gradient methods and overall pretraining efficiency. In contrast, we operate in the supervised reasoning SFT stage. Our reweighting signal is not the loss itself but the loss reduction across epochs ($\Delta$ loss), which measures how much additional learning each example continues to provide. This design is motivated by our empirical finding, across multiple base models and datasets, that training on negative reasoning samples improves OOD performance, and our method is explicitly built to exploit negative reasoning trajectories to enhance OOD generalization. Conceptually, we adopt the general idea of online sample reweighting from [1], but instantiate it in a setting that is closer to practical SFT rather than generic pretraining and center it on negative examples and training dynamics. Accordingly, [1] develops global convergence results for convex objectives, whereas our theory studies improvements in conditioning and generalization for non-convex LLMs via the spectrum of the empirical Fisher and stability bounds.
>
> The other issues regarding the formulas that you raised in Weakness 1 have been fully addressed in Appendix A.2.
>
> [1] https://openreview.net/forum?id=gU4ZgQNsOC

---

> ### Author Response · Authors · 2025-11-21
> **Official Comment by Authors (2)**
>
> # Weakness 2
> We thank the reviewer for suggesting a comparison with [2] and [3]. At a high level, GLOW focuses on improving out-of-domain generalization for LLM reasoning during supervised fine-tuning, whereas [2] and [3] target different problems and use different reweighting criteria.
> From the problem perspective, [2] studies self-improvement from self-generated data, using importance weighting to filter and reweight noisy model-generated samples so that the model can learn safely from its own outputs. [3] is a theoretical active learning work that analyzes convergence of SGD when examples are sampled based on loss or uncertainty, primarily for convex or well-behaved settings and generic classifiers. In contrast, GLOW is designed around negative chain-of-thought trajectories in LLMs and aims specifically to strengthen OOD reasoning performance during SFT, rather than to design a general data acquisition policy or a generic self-improvement filter.
> From the method perspective, [2] uses an importance weight that approximates distribution shift to reweight self-generated samples, and [3] analyzes uncertainty-based sampling and proposes an adaptive step-size rule. GLOW instead reweights examples according to their loss reduction across epochs and uses this dynamic “gain” signal to decide when negative reasoning samples should be emphasized. Thus, GLOW is instantiated and validated in the context of LLM reasoning, but the underlying idea of using loss-reduction–based reweighting is in principle applicable to other supervised tasks, which we see as an interesting direction beyond the current work.
>
> [2] https://arxiv.org/abs/2408.09849
> [3] https://openreview.net/forum?id=GLUIuli3Sm

---

> ### Author Response · Authors · 2025-11-21
> **Official Comment by Authors (3)**
>
> # Q1 & Q2
>
> We thank the reviewer for raising this point and for highlighting [4]. Intuitively, one might expect that training on incorrect samples, especially in an RL-style setup where they are explicitly penalized, would bias the model toward undesirable behavior. Our experiments consistently contradict this intuition: emphasizing negative reasoning samples during SFT improves out-of-distribution (OOD) reasoning performance across many models and datasets (although slightly worse than that of positive-only training). We view this counter-intuitive effect as one of the main empirical findings of our work.
>
> First, [4] is an RLVR-style method: the model is updated with policy gradients to maximize expected reward, and incorrect trajectories are explicitly punished through negative advantage. In contrast, GLOW remains in the SFT regime with a standard cross-entropy objective on logged trajectories. We only apply a deterministic reweighting of the loss, and the optimization target stays “match the training distribution of CoT trajectories.” As a result, negative samples in GLOW act as supervised signals within the same modeling objective, rather than as repulsive signals as in RL.
>
> Second, after an in-depth analysis, we believe the empirical gains from negative samples are linked to their structure and diversity. Our case study in Fig. 8 shows that many negative trajectories still contain clear, well-organized reasoning steps; they are “wrong” only at the final answer, so they continue to teach task format and step-wise reasoning instead of pure noise. As shown in Fig. 1, SFT on negative samples produces slower but smoother loss curves and leads to better OOD performance than positive-only SFT, suggesting that these trajectories act as a regularizer: they prevent the model from overfitting to a narrow set of shortcut-style correct solutions. From an entropy perspective, Table 5 further shows that negative-based training preserves higher predictive entropy over candidate reasoning paths than positive-only SFT, so the model maintains a richer set of plausible solutions instead of collapsing onto a few over-confident patterns.
>
> Overall, our results indicate that, under GLOW’s $\Delta$ loss-based reweighting within SFT, negative reasoning samples do not bias the model toward systematically incorrect answers. Instead, their structural coherence and diversity provide useful supervision that improves robustness and OOD generalization.
>
> # Q3
> IRM provides a theoretical framework for understanding why diverse environments improve generalization. We interpret “diverse erroneous reasoning patterns” as multiple environments: the model is required to perform well across these environments, which encourages it to learn more stable reasoning representations. In our setting, an environment corresponds to a distinct type of incorrect reasoning; these error types induce different distributional conditions, that is, different local data distributions from the perspective of the language model. Concretely, in our formulation, $\Phi$ denotes the hidden-state mapping that sends $(x, y_{<t})$ to $h_t$, and $w$ denotes the shared token-level classifier in the output layer. When the environments are diverse, the model must learn reasoning representations that remain stable across error patterns so that the shared head $w$ can perform reliably in all environments.
>
> # Q4
> Thank you for pointing it out. The blue and red curves come from two completely independent experiments. We sampled the same number of positive and negative samples and trained Qwen2.5-32B separately on each. The blue curve shows the loss when training only on negative samples, and the red curve shows the loss when training only on positive samples.

---

> ### Author Response · Authors · 2025-11-21
> **Official Comment by Authors (4)**
>
> # Q5
> We thank the reviewer for this thoughtful question. In our view, GLOW and REINFORCE-style methods such as [1] address different levels of the training pipeline and are conceptually compatible rather than in conflict.
> From the objective perspective, [1] is a reinforcement learning method. It updates the model by policy gradients, scales gradients by a reward (or advantage), and uses negative rewards to push probability mass away from bad trajectories. GLOW stays strictly in the supervised fine-tuning regime. We optimize a standard cross-entropy loss on logged reasoning trajectories and only reweight this loss across examples based on their loss reduction. Negative samples are not “punished”; they receive higher or lower weight depending on whether they continue to provide useful learning signal.
> In STaR-style self-improvement, the verifier first selects trajectories and then uses them to form a supervised dataset. GLOW can be applied exactly at this SFT stage: once the STaR dataset is fixed, one can use Δloss-based reweighting over the selected trajectories, while any additional RL phase can still rely on reward indicators as usual. We have not explored this combination empirically, but conceptually GLOW is a supervised reweighting mechanism on CoTs, whereas [1] is a reward-driven policy optimization method, and the two are complementary rather than contradictory.
>
> [1] https://openreview.net/forum?id=ftVlLG9cks
>
> # Q6
> We use the cosine learning rate scheduler (lr_scheduler_type = cosine in LLaMA-Factory) and record the loss every 10 steps during training, then plot the raw, unsmoothed loss curves. We observe the same pattern across multiple datasets and base models (see Fig. 8). Based on this evidence, we hypothesize that the behavioral difference between positive and negative samples largely comes from how new examples are exposed across epochs. Positive samples are memorized more quickly and show sharper loss drops between epochs, whereas negative samples continue to produce stable loss reductions over more epochs, which is associated with better OOD generalization.
>
> # Q7
> Your question touches on a very practical consideration. When compute resources are sufficient, using full chain-of-thought is generally the preferred choice for equipping LLMs with reasoning ability. For example, this is the case in methods such as STaR, as well as in the Qwen models distilled from DeepSeek-R1 released by the authors.
>
> # Q8
> We appreciate the reviewer’s question. Our experiments do support the claim that “training only on negative samples can outperform training only on positive samples,” but this holds under two explicit conditions. First, the gains are observed in out-of-domain evaluation. Across multiple base models and datasets, negative-only SFT consistently achieves higher OOD performance than positive-only SFT (see Table 1 and Table 2). Second, this effect relies on the typical structure of reasoning data, where negative trajectories remain informative format and semantically diverse, and differ from positives mainly in the final answer. Under these two assumptions, our results indicate that negative-only training can indeed be more effective than positive-only training for OOD reasoning.
>
> # Q9
> We are aware of this finding as well. In our setting, we do not expect this issue to arise. In RL, the problem typically comes from using vLLM for inference while training with FSDP + PyTorch/DeepSpeed. Different kernels and parallelization strategies introduce rounding errors that accumulate during autoregressive sampling, so the rollout policy μ and the policy π used to compute log π during training can diverge numerically. In SFT, by contrast, we train on pre-collected data and run both forward and backward passes within a single, consistent system, which avoids this mismatch.

---

> ### Author Response · Authors · 2025-11-25
> **Request for Feedback Before Discussion Period Ends**
>
> Dear Reviewer 4qLV:
>
> As the author-reviewer discussion period approaches its end, we kindly request your feedback on our rebuttal. Your insights on whether we have effectively addressed your concerns would be greatly appreciated.
>
> We are truly grateful for the time and expertise you have dedicated to reviewing our work. Your thoughtful comments and suggestions have significantly contributed to improving the quality of our research.
>
> Thank you once again for your valuable feedback. We look forward to your guidance and hope to address any remaining concerns before the discussion period closes.
>
> Yours Sincerely,
>
> All Authors

---

> ### Author Response · Authors · 2025-11-27
> **If there are additional questions that need clarification, please feel free to ask and discuss.**
>
> Dear Reviewer 4qLV:
>
> Thank you for your meticulous review, which played a pivotal role in enhancing the quality of our paper.
>
> We have responded to the concerns highlighted in your review. If you require further clarification or have additional questions regarding my responses, please feel free to submit them through the official comment button.
>
> Your dedicated efforts in reviewing our paper are greatly appreciated!
>
> Best Regards,
>
> Authors

---

> ### Comment · Reviewer_4qLV · 2025-11-27
>
> Apologies for my late response, and I thank the authors for their detailed response.
>
> I have some remaining concerns as follows, mostly at theoretical and conceptual level:
> - I understand the role of energy condition, but this still an assumption that is not verified rigorously. What I mean is, in Assumption A.1, (A1)-(A4) are all standard and reasonable, but (A5) seems like the assumption that directly leads to the conclusion, i.e., the theorem statement hinges on (A5) in a rather trivial manner. This seems to be a quite strong assumption as this must hold for all iterations. Thus, I'm not convinced on the current theoretical framework that the paper proposes.
> - Are the authors claiming that their algorithm is resulting in better gradient directions in the *original* loss $\sum_i \ell_i^{(t)}$? Or are the authors claiming that the newly weighted loss $\sum_i w_i^{(t)} \ell_i^{(t)}$ becomes more benign, *and* retains all the favorable properties (e.g., high ood accuracy, etc)?
> - It may be interesting to expand on Figure 3 to a figure that sequentially shows the weight distribution (histogram) at certain checkpoints. The phrase "large weight" is quite imprecise. Or maybe take specific samples and plot the evolution of its weight? Because each iteration's weight is history-dependent (I think Markovian to be precise, as it depends on previous epoch's losses), the optimization process is quite correlated compared to vanilla optimization process.
>
> Despite this, majority of my concerns have been addressed, and considering the empirical impact of the paper's claims as well as authors' newly added experimental results during the rebuttal, I raise my score from 4 to 6 and confidence from 2 to 3.

---

> ### Author Response · Authors · 2025-11-30
> **Official Comment by Authors (5)**
>
> # Q1
> Thank you for raising this concern. In the revised PDF, we have further corrected Assumption A5. Our intended meaning is not that the conclusion is assumed to hold trivially at every training step; rather, A5 is a stylized mechanism-level assumption that captures the effect we aim to model—our reweighting upweights low-gain samples and, on average, injects curvature into under-explored directions, which then enables the subsequent conditioning/stability analysis. After the revision, the assumption more clearly emphasizes this conditional and stochastic nature (and the role of ($U_t$) as a local low-curvature subspace), making the theoretical statement explicitly a conditional guarantee about the geometry of the reweighted objective rather than a blanket claim about the entire optimization trajectory.
>
> # Q2
> Thank you for this important clarification request. Our theoretical analysis is about the **reweighted surrogate objective** that our algorithm actually optimizes:
>
> $$
> R_w^{(t)}(\theta)=\frac{1}{N}\sum_i w_i^{(t)}\,\ell_i(\theta).
> $$
>
> We study how the induced weighting changes the geometry and stability of $R_w^{(t)}$ (e.g., Fisher/Hessian conditioning in low-curvature directions) under the stated assumptions. Importantly, while training uses dynamic reweighting, all reported performance (including OOD) is evaluated using the **standard, original evaluation protocol** on held-out benchmarks (i.e., the same metrics as vanilla ERM). Thus, our claims about **high OOD accuracy** are empirical: optimizing this reweighted ERM yields models that generalize better under the original evaluation objectives.
>
> # Q3
> Thank you for the suggestion. We have revised Figure 3 and the corresponding text to make the definition precise. At each optimization step, we select the example with the largest upweighting signal. We then calculate, for each epoch, the fraction of negatives among these stepwise-selected examples. With this clarified definition, Figure 3 still supports the same takeaway—negatives are consistently over-represented among the most strongly upweighted examples (especially early in training), and the ratio gradually decreases as training progresses. We also agree that the weights are history-dependent; we now describe this explicitly in the text, and Figure 3 is intended to summarize the aggregate effect of this correlated reweighting dynamics.

---

### Author Response · Authors · 2025-11-21
**Official Comment by Authors**

We thank all reviewers for their careful reading of our paper and for the constructive feedback. We have updated the paper in this revised version and provided point-by-point responses that address each reviewer’s concerns, weaknesses, and questions in detail. We hope that these responses and new results help resolve the issues raised. If any points remain unclear, we would be very happy to continue the discussion and we look forward to an active and constructive exchange.

---

### Author Response · Authors · 2025-11-30
**Key Takeaways and Contributions**

In summary, our work shows that negative distilled reasoning trajectories provide a strong and transferable training signal. Using these trajectories for SFT to build reasoning ability improves out-of-domain generalization under standard evaluation.

* We surprisingly find that training on negative reasoning trajectories delivers more consistent out-of-domain gains than training on positive reasoning trajectories alone.
* We analyze negative trajectories from both the data and training perspectives, and provide converging evidence from multiple diagnostic views.
* We propose GLOW, a method derived from these observations. It improves overall performance on both in-domain and out-of-domain benchmarks. It does not require prior rejection sampling, which improves data efficiency. It also outperforms standard SFT on the full dataset.

Overall, the paper empirically demonstrates that learning from negative trajectories yields stronger out-of-domain generalization. It also introduces GLOW, which improves performance across diverse domains. Moreover, it provides a stronger initialization for subsequent RL stage.

---

### Author Response · Authors · 2025-12-02
**Response Summary for Area Chair**

Dear Area Chair,


Thank you for overseeing the review process of our paper and for the time and effort you and the reviewers have invested. We appreciate the constructive discussion throughout the rebuttal.


We would also like to highlight a key development in the discussion trajectory: **the reviewer 4qLV has updated their score from 4 to 6, and the reviewer SmfD updated their score from 2 to 4 after the first-round discussion. In the second round, we added new experiments and analyses (summarized below) to directly address reviewer SmfD's new concerns**. Most notably, (i) RL compatibility/initialization results, and (ii) test-time scaling (pass@k) evidence that strengthens the core empirical claim. Then the API leakage occurred, which disrupted our further communication with the reviewers.


We have addressed concerns point-by-point, and all revisions are marked in blue in the revised PDF for ease of inspection. Below we summarize the major revisions and then provide short reviewer-by-reviewer highlights.

---

> ### Author Response · Authors · 2025-12-02
> **Major Revisions**
>
> 1. **Clarified scope and framing of the paper.** We revised the abstract/introduction to make explicit that we study *CoT-based reasoning SFT*, and that the commonly used recipe involves distilled CoTs with reject sampling filtering, our mechanisms and claims are scoped to this setting.
>
> 2. **Strengthened and clarified the theory presentation (Eq. (7) and proof details).** We expanded explanations around the GLOW objective, clarified notation, and corrected a typo in the PSD increment term. We also clarified the role of the low-curvature subspace $U$ and the corresponding energy condition used in the proof sketch.
>
> 3. **Improved positioning versus related reweighting work.** We added a clearer comparison to loss-based reweighting in pretraining, importance-weighted self-improvement, and uncertainty-based sampling theory, emphasizing the different setting (reasoning SFT), signal (cross-epoch $\Delta$-loss), and goal (OOD reasoning generalization).
>
> 4. **Added new experiments answering the RL-stage practical-impact concern.** During rebuttal we ran controlled RL experiments with *verl* on Qwen2.5-7B-base (GSM8K), comparing **SFT vs. GLOW** and the corresponding **SFT+RL vs. GLOW+RL** settings while keeping RL details identical. Results show the gains from GLOW persist after RL and provide a stronger initialization.
>
> 5. **Added test-time scaling evidence via pass@k.** To address concerns that single-sample accuracy may be an imperfect proxy, we report pass@4/8/16/32 on Math500 and Minerva, showing consistent advantages for models trained with negative trajectories across k on general reasoning dataset.
>
> 6. **Added learning-dynamics evidence that negatives are not noise.** We added checkpoint-based evaluations (e.g., 5/10/15/20 epochs) comparing positive vs. negative distilled trajectories to demonstrate consistent progress patterns and complementary evidence for meaningful supervision.
>
> 7. **Clarified presentation points and minor fixes.** We improved wording where reviewers found ambiguity and clarified figure annotations.

---

> ### Author Response · Authors · 2025-12-02
> **Briefly Summarized Response to Reviewer 4qLV (score 4 → updated to 6.)**
>
> - **Theory clarity & corrected notation:**
>
>   -  We clarified Eq. (7) and the meaning of all symbols, and corrected the typo in the proof (the PSD increment uses $\Delta w_i\, g_i g_i^\top$, not $\delta w_i\, g_i g_i^\top$). Remaining formula-level issues raised in Weakness 1 are addressed in Appendix A.2.
>
>   - Our theoretical goal is to show that reweighting improves conditioning **specifically along ill-conditioned directions** (small eigenvalues). We therefore introduce the subspace $U$ to isolate these low-curvature directions, and add an energy condition to formalize that the reweighting injects sufficient gradient "mass" into $U$, so that curvature along $U$ is provably enhanced.
>
>   - We clarified that the analysis is on the **reweighted surrogate objective**
>   $
>   R_w^{(t)}(\theta)=\frac{1}{N}\sum_i w_i^{(t)}\,\ell_i(\theta),
>   $
>   studying how dynamic $\Delta$-loss-based weighting changes geometry/stability under a stylized mechanism-level assumption. In contrast, all OOD performance claims are empirical and evaluated under the standard benchmark protocol (same as vanilla ERM), i.e., the theory explains a plausible mechanism while the accuracy gains are validated experimentally.
>
> - **Relation to prior work on reweighting / self-improvement / active learning:**
>   We consolidated the distinction from prior work: unlike loss-based pretraining reweighting that uses instantaneous loss and targets convergence/efficiency under convex-style assumptions, **GLOW operates in reasoning SFT and uses cross-epoch loss reduction ($\Delta$-loss / gain)** to emphasize samples that keep providing learning signal, with theory focused on conditioning and stability in LLM training. We also clarified differences from importance-weighted self-improvement (distribution-shift correction for self-generated data) and active-learning theory (sampling policies in well-behaved regimes).
>
> [1] https://openreview.net/forum?id=gU4ZgQNsOC
>
> [2] https://arxiv.org/abs/2408.09849
>
> [3] https://openreview.net/forum?id=GLUIuli3Sm
>
> [4] https://openreview.net/forum?id=ftVlLG9cks
>
> - **Why negative trajectories help under SFT (and why this differs from RL "punishment"):**
>   We emphasized that RLVR-style methods explicitly penalize incorrect trajectories via negative advantage, whereas GLOW stays strictly in SFT with a standard cross-entropy objective on logged CoTs. **Negative trajectories are not treated as repulsive signals, instead, they are selectively emphasized through our $\Delta$-loss reweighting, which strengthens the training signal from those negatives that continue to yield meaningful learning progress.** Empirically, many "negative" CoTs remain well-structured and are often incorrect mainly at the final answer, so they still provide meaningful supervision on intermediate-step structure and reasoning format. Moreover, because negative trajectories exhibit more diverse reasoning patterns, emphasizing them can reduce over-concentration on a narrow set of high-confidence solution patterns learned from positives alone, thereby improving robustness and OOD generalization (supported by our case studies, the observed loss-curve dynamics, and the higher predictive entropy compared to positive-only SFT).
>
>
> - **Figure caption clarifications:**
>   We clarified that the red and blue loss curves come from two independent runs (positive-only vs. negative-only) with the same sampling budget, plotted using raw unsmoothed losses under a cosine LR schedule. We also clarified the definition behind the "most-upweighted examples" statistic (Fig. 3) and noted the history-dependent nature of weights.

---

> ### Author Response · Authors · 2025-12-02
> **Briefly Summarized Response to Reviewer ZzKP**
>
> - **Difference from "negative reinforcement" in RL:** We clarified that RL-style negative reinforcement uses incorrect trajectories as *repulsive signals* under reward maximization, whereas our method stays strictly in SFT and treats negative trajectories as additional supervision examples whose weight is determined by learning dynamics $\Delta$-loss, not correctness.
> - **Generality of takeaways:** We emphasized that improvements are primarily driven by the *diversity of errors* (broader trajectory coverage), rather than any single mistake category, and we clarified the intended scope/limits of this claim.

---

> ### Author Response · Authors · 2025-12-02
> **Briefly Summarized Response to Reviewer SmfD (score 2 → updated to 4 after round 1.)**
>
> # Round 1 — Key Resolutions & Added Evidence
>
> - ****Practical relevance of SFT-stage generalization (vs. "RL generalizes, SFT memorizes"):****
>   We clarified our intent: this work is an **SFT-level** study on whether negative CoT trajectories can be exploited as useful supervision to improve **cross-domain generalization before RL**. We argued (with practical rationale) that stronger SFT initialization can benefit downstream RL by improving the starting policy’s reasoning breadth and exploration quality, while a full RL-transfer study is beyond the original scope (later addressed with new RL experiments in Round 2).
>
> - ****Scope clarification (abstract + terminology):****
>   We revised the abstract to explicitly scope the discussion to **CoT-based reasoning SFT** (not all SFT), and clarified that "only correct CoTs" refers to the common correctness-filtered reasoning post-training recipe. We also rewrote confusing phrases such as "dominant SFT" and clarified why distillation is mentioned (as the standard source of reasoning trajectories in many pipelines).
>
> - ****Evaluation/communication fixes (OOD definition, figures, tables):****
>   We corrected "commonsense benchmark" → "general reasoning benchmarks", clarified why MMLU-math is still distributionally OOD relative to AoPS-style long-form math-CoT training, fixed unclear "+X" annotations in Fig. 1 by stating they denote **absolute improvements over the pre-SFT baseline**, and updated table captions to explain color/bold coding so the tables are self-contained.
>
> - ****Prove "negatives are richer signals, not noise":****
>   We clarified that loss-curve monotonicity is only a sanity check (MLE loss drops), and the real evidence comes from generalization tables. For "richer reasoning patterns", we pointed to **(i) error-type diversity analysis on negatives (taxonomy-based) and (ii) higher policy entropy under negative training** (evidence of broader reasoning-path distribution), which together support the claim that negatives provide transferable supervision signals.
>
> - ****Added experiments to address "missing backbone models in Sec. 5" (new tables):****
>   To ensure the Sec. 5 phenomena are not backbone-specific, we added **Qwen2.5-32B** and **Llama3.1-8B** results (SFT vs. GLOW) under both MATH and MMLU training data. GLOW yields consistent gains across **model families** (Qwen vs. Llama) and **scales** (8B → 32B), strengthening the robustness of the conclusions.
>
> |             | Training Data | Method |  math500  |  minerva  |  olympia  |    amc    |  Avg_math |    MMLU   |  MMLU-Pro |    BBH    | Avg_general |  ACPBench |   HeadQA  | Avg_other |
> |:-----------:|:-------------:|:------:|:---------:|:---------:|:---------:|:---------:|:---------:|:---------:|:---------:|:---------:|:-----------:|:---------:|:---------:|:---------:|
> | llama3.1-8B |      MATH     |   SFT  |   41.20   |   18.01   |   14.67   |   15.00   |   22.22   |   62.48   |   36.88   |   55.12   |    51.49    |   32.96   |   65.90   |   49.43   |
> |             |               |  GLOW  | **44.60** | **20.59** | **15.11** | **17.50** | **24.45** | **63.80** | **38.34** | **58.17** |  **53.44**  | **35.04** | **66.70** | **50.87** |
> |             |      MMLU     |   SFT  |   20.00   |   15.81   |    6.52   |    2.50   |   11.21   |   66.49   |   40.56   |   53.73   |    53.59    |   36.06   |   69.55   |   52.81   |
> |             |               |  GLOW  | **24.80** | **20.59** |  **6.96** | **12.50** | **16.21** | **68.52** | **42.95** | **57.53** |  **56.33**  | **39.72** | **72.57** | **56.15** |
> | Qwen2.5-32B |      MATH     |   SFT  |   92.20   |   52.57   |   57.19   |   85.00   |   71.74   |   85.22   |   73.10   |   83.53   |    80.62    |   50.67   |   84.90   |   67.79   |
> |             |               |  GLOW  | **93.40** | **54.41** | **59.11** | **92.50** | **74.86** | **85.51** | **74.14** | **83.98** |  **81.21**  | **51.97** | **85.19** | **68.58** |
> |             |      MMLU     |   SFT  |   86.60   |   46.69   |   55.70   |   80.00   |   67.25   |   79.06   |   61.15   |   79.94   |    73.38    |   49.89   |   83.01   |   66.45   |
> |             |               |  GLOW  | **89.00** | **47.06** | **58.67** | **82.50** | **69.31** | **80.81** | **64.72** | **81.98** |  **75.84**  | **52.08** | **83.73** | **67.91** |
>
> - ****Limitations:****
>   We added a limitation section in the appendix.

---

> > ### Author Response · Authors · 2025-12-02
> > **Briefly Summarized Response to Reviewer SmfD (Round 2 concerns addressed with new experiments.)-1**
> >
> > # Round 2-1
> >
> > - **Directly addressing "Does SFT-stage gain transfer to RL?" (new RL experiments with verl):**
> >   We added controlled RL experiments to test whether the SFT-stage improvement provided by GLOW remains beneficial after an identical RL pipeline. We fine-tuned Qwen2.5-7B-base on GSM8K under four settings (SFT, SFT+RL, GLOW, GLOW+RL), keeping all RL hyperparameters/data/optimization details fixed and only changing whether the preceding SFT used GLOW reweighting. **GLOW improves OOD metrics before RL, and GLOW+RL further improves OOD metrics after RL compared to SFT+RL**, indicating GLOW provides a stronger initialization that remains beneficial under RL-style post-training.
> >
> > |            |         | GSM8K(test) | Math500  | Minerva  | Olympia  | AMC  |  MMLU | MMLU-Pro |
> > |------------|---------|-------------|:-------:|:--------:|:--------:|:--------:|:-----:|:--------:|
> > | Qwen2.5-7B | SFT     | 89.20       | 76.60   | 40.07    | 38.96    | 55.00    | 72.24 | 53.71    |
> > |            | SFT+RL  | 92.58       | 78.20   | 40.24    | 39.67    | 52.50    | 72.82 | 53.95    |
> > |            | GLOW    | 89.80       | 79.60   | 40.07    | 41.04    | **60.00**| 73.99 | 55.77    |
> > |            | GLOW+RL | **93.06**   | **80.20**| **42.28**| **41.78**| 57.50    | **76.47** | **57.37** |
> >
> > - **Addressing "test-time scaling / pass@k" (new pass@k results):**
> >   We added pass@4/8/16/32 comparisons (Math500, Minerva) between models trained with positive vs. negative samples. Negative-trained models consistently outperform positive-trained models across all $k$ values on OOD benchmarks, strengthening the claim beyond single-sample decoding and supporting the **"broader exploration and stronger base policy for RL"** interpretation.
> >
> > ### Math500 (Pass@k)
> > | Model     | Pass@4 | Pass@8 | Pass@16 | Pass@32 |
> > |-----------|:------:|:------:|:-------:|:-------:|
> > | positive  | 85.38  | 88.41  | 90.56   | 92.00   |
> > | negative  | 87.60  | 89.61  | 91.65   | 93.20   |
> >
> > ### Minerva (Pass@k)
> > | Model     | Pass@4 | Pass@8 | Pass@16 | Pass@32 |
> > |-----------|:------:|:------:|:-------:|:-------:|
> > | positive  | 47.78  | 52.35  | 56.44   | 60.29   |
> > | negative  | 49.40  | 54.53  | 59.16   | 62.87   |

---

> ### Author Response · Authors · 2025-12-02
> **Briefly Summarized Response to Reviewer SmfD (Round 2 concerns addressed with new experiments.)-2**
>
> # Round 2-2
>
> - ****Addressing "negatives might be noise" with learning-dynamics evidence (new checkpoint evaluations):****
>   We added checkpoint evaluations at 5/10/15/20 epochs for both Qwen2.5-7B and Qwen2.5-32B, under math reasoning and general reasoning dataset training, each comparing positive vs. negative distilled trajectories. **Across epochs, negative training consistently yields stronger OOD performance, demonstrating that negatives provide transferable training signals rather than merely fitting noise.**
>
> ### Qwen2.5-7B — **Math reasoning** (Positive vs. Negative distilled trajectories)
> | Epoch  | Trajectory | Math500 | Minerva | Olympia |  AMC  |  MMLU | MMLU-Pro |  BBH  |
> |-------:|-----------:|:-------:|:-------:|:-------:|:-----:|:-----:|:--------:|:-----:|
> | Base   | —          | 58.40   | 26.84   | 26.07   | 52.50 | 55.80 | 26.56    | 51.10 |
> | 5epoch | Positive   | 72.80   | 37.13   | 37.19   | 45.00 | 60.95 | 30.34    | 54.69 |
> |        | Negative   | 71.20   | 31.99   | 31.56   | 47.50 | **62.58** | **44.04** | **56.28** |
> | 10epoch| Positive   | 75.80   | 38.24   | 40.59   | 65.00 | 64.06 | 32.50    | 61.62 |
> |        | Negative   | 77.20   | 34.93   | 39.26   | 50.00 | **71.39** | **52.14** | **69.49** |
> | 15epoch| Positive   | 77.20   | 36.76   | 41.93   | 55.00 | 60.81 | 32.15    | 59.69 |
> |        | Negative   | 78.60   | 39.71   | 38.37   | 52.50 | **72.10** | **52.24** | **71.09** |
> | 20epoch| Positive   | 78.00   | 36.76   | 41.78   | 57.50 | 61.03 | 32.70    | 60.58 |
> |        | Negative   | 77.60   | 40.44   | 38.37   | 57.50 | **73.11** | **53.74** | **71.73** |
>
> ### Qwen2.5-7B — **General reasoning** (Positive vs. Negative distilled trajectories)
> | Epoch  | Trajectory | Math500 | Minerva | Olympia |  AMC  |  MMLU | MMLU-Pro |  BBH  |
> |-------:|-----------:|:-------:|:-------:|:-------:|:-----:|:-----:|:--------:|:-----:|
> | Base   | —          | 58.40   | 26.84   | 26.07   | 52.50 | 55.80 | 26.56    | 51.10 |
> | 5epoch | Positive   | 72.00   | 36.76   | 37.33   | 47.50 | 73.62 | 50.61    | 64.05 |
> |        | Negative   | **76.80** | **36.76** | **37.78** | **47.50** | 71.09 | 43.99 | 66.00 |
> | 10epoch| Positive   | 74.60   | 37.50   | 41.48   | 55.00 | 73.79 | 53.32    | 69.73 |
> |        | Negative   | **76.80** | **37.87** | 40.30 | 52.50 | 71.43 | 45.87 | 68.84 |
> | 15epoch| Positive   | 72.00   | 37.50   | 39.26   | 50.00 | 74.11 | 53.91    | 68.34 |
> |        | Negative   | **76.80** | **37.13** | **41.48** | **55.00** | 71.30 | 44.62 | 69.30 |
> | 20epoch| Positive   | 74.40   | 37.50   | 39.85   | 50.00 | 73.42 | 53.22    | 68.23 |
> |        | Negative   | **77.00** | 37.13 | **42.07** | **60.00** | 71.23 | 45.79 | 69.46 |
>
> ### Qwen2.5-32B — **Math reasoning** (Positive vs. Negative distilled trajectories)
> | Epoch  | Trajectory | Math500 | Minerva | Olympia |  AMC  |  MMLU | MMLU-Pro |  BBH  |
> |-------:|-----------:|:-------:|:-------:|:-------:|:-----:|:-----:|:--------:|:-----:|
> | Base   | —          | 63.20   | 34.19   | 26.52   | 35.00 | 68.34 | 39.80    | 58.65 |
> | 5epoch | Positive   | 90.20   | 49.63   | 59.11   | 85.00 | 76.53 | 46.77    | 78.04 |
> |        | Negative   | 88.40   | 45.22   | 52.30   | 85.00 | **83.07** | **68.23** | **83.55** |
> | 10epoch| Positive   | 92.60   | 50.00   | 60.44   | 85.00 | 78.63 | 51.67    | 79.01 |
> |        | Negative   | 92.20   | 51.10   | 57.93   | 85.00 | **85.14** | **73.75** | **84.22** |
> | 15epoch| Positive   | 93.00   | 48.53   | 62.07   | 90.00 | 78.72 | 51.99    | 80.57 |
> |        | Negative   | 91.20   | 50.74   | 57.33   | 90.00 | **85.02** | **73.48** | **84.62** |
> | 20epoch| Positive   | 91.40   | 50.74   | 60.89   | 85.00 | 79.01 | 54.31    | 80.61 |
> |        | Negative   | 92.20   | 50.74   | 58.37   | 95.00 | **85.47** | **73.53** | **84.51** |
>
> ### Qwen2.5-32B — **General reasoning** (Positive vs. Negative distilled trajectories)
> | Epoch  | Trajectory | Math500 | Minerva | Olympia |  AMC  |  MMLU | MMLU-Pro |  BBH  |
> |-------:|-----------:|:-------:|:-------:|:-------:|:-----:|:-----:|:--------:|:-----:|
> | Base   | —          | 63.20   | 34.19   | 26.52   | 35.00 | 68.34 | 39.80    | 58.65 |
> | 5epoch | Positive   | 84.60   | 44.85   | 52.00   | 62.50 | 82.10 | 66.54    | 80.03 |
> |        | Negative   | **85.00** | 44.49 | 51.26 | **77.50** | 78.74 | 57.48 | 79.09 |
> | 10epoch| Positive   | 86.60   | 46.69   | 55.70   | 75.00 | 81.14 | 67.01    | 80.69 |
> |        | Negative   | **87.20** | 46.30 | 54.52 | **75.00** | 79.01 | 60.43 | 80.88 |
> | 15epoch| Positive   | 85.00   | 47.06   | 56.59   | 75.00 | 81.73 | 68.33    | 81.73 |
> |        | Negative   | **86.40** | **47.79** | 55.70 | 65.00 | 77.77 | 57.14 | 79.97 |
> | 20epoch| Positive   | 85.20   | 46.69   | 56.15   | 75.00 | 81.97 | 68.54    | 81.60 |
> |        | Negative   | **86.40** | **47.06** | **56.89** | 72.50 | 77.99 | 58.34 | 80.71 |

---

> > ### Author Response · Authors · 2025-12-02
> > **Briefly Summarized Response to Reviewer SmfD (Round 2 concerns addressed with new experiments.)-3**
> >
> > # Round 2-3
> >
> > - **Distillation practicality (clarification):**
> >   We explicitly stated that distillation is not a practical limitation. It is a simple, scalable, and effective way to obtain both positive and negative trajectories with consistent formatting, while emphasizing that GLOW itself depends on per-sample $\Delta$-loss dynamics rather than a hard sample-type split.
> >
> > - **Overall conclusion with respect to the key concern:**
> >   The newly added RL table + pass@k + checkpoint analyses jointly address the reviewer’s practical-impact concerns by showing:
> >   (i) **SFT gains transfer and persist under RL** (improved initialization),
> >   (ii) **improvements are not an artifact of single-sample decoding** (pass@k),
> >   (iii) **negatives trajectories provide stable, transferable learning signals throughout training** (checkpoint trends).

---

### Note · Authors · 2026-01-04

I have read and agree with the venue's withdrawal policy on behalf of myself and my co-authors.